# Diffusion-based Extreme Image Compression with Compressed Feature Initialization

## Abstract

Diffusion-based extreme image compression methods have achieved impressive performance at extremely low bitrates. However, constrained by the iterative denoising process that starts from pure noise, these methods are limited in both fidelity and efficiency. To address these two issues, we present **R**elay **R**esidual **D**iffusion **E**xtreme **I**mage **C**ompression (**RDEIC**), which leverages compressed feature initialization and residual diffusion. Specifically, we first use the compressed latent features of the image with added noise, instead of pure noise, as the starting point to eliminate the unnecessary initial stages of the denoising process. Second, we design a novel relay residual diffusion that reconstructs the raw image by iteratively removing the added noise and the residual between the compressed and target latent features. Notably, our relay residual diffusion network seamlessly integrates pre-trained stable diffusion to leverage its robust generative capability for high-quality reconstruction. Third, we propose a fixed-step fine-tuning strategy to eliminate the discrepancy between the training and inference phases, further improving the reconstruction quality. Extensive experiments demonstrate that the proposed RDEIC achieves state-of-the-art visual quality and outperforms existing diffusion-based extreme image compression methods in both fidelity and efficiency. The source code and pre-trained models will be released.

## 1 Introduction

Extreme image compression is becoming increasingly important with the growing demand for efficient storage and transmission of images where storage capacity or bandwidth is limited, such as in satellite communications and mobile devices. Conventional compression standards like JPEG (Wallace, 1991), BPG (Bellard, 2014) and VVC (Bross et al., 2021) rely on hand-crafted rules and block-based redundancy removal techniques, leading to severe blurring and blocking artifacts at low bitrates. Hence, there is an urgent need to explore extreme image compression methods.

In recent years, learned image compression methods have attracted significant interest, outperforming conventional codecs. However, distortion-oriented learned compression methods (Xie et al., 2021; Zhu et al., 2021; Liu et al., 2023; Li et al., 2024a) optimize for the rate-distortion function alone, resulting in unrealistic reconstructions at low bitrates, typically manifested as blurring or over-smoothing. Perceptual-oriented learned compression methods (Agustsson et al., 2019; Mentzer et al., 2020; Muckley et al., 2023; Yang & Mandt, 2023) introduce generative models, such as generative adversarial networks (GANs) (Goodfellow et al., 2014) and diffusion models (Ho et al., 2020), to enhance the perceptual quality of reconstructions. However, these methods are optimized for medium to high bitrates instead of extremely low bitrates such as below 0.1 bpp. As a result, these methods experience significant quality degradation when the compression ratio is increased.

Recently, diffusion-based extreme image compression methods (Lei et al., 2023; Careil et al., 2024; Li et al., 2024b) leverage the robust generative ability of pre-trained text-to-image (T2I) diffusion models, achieving superior visual quality at extremely low bitrates. Nonetheless, these methods are constrained by the inherent characteristics of diffusion models. Firstly, these methods rely on an iterative denoising process to reconstruct raw images from pure noise, which is inefficient for inference (Li et al., 2024b). Secondly, initiating the denoising process from pure noise introduces significant randomness, compromising the fidelity of the reconstructions (Careil et al., 2024). Thirdly, there is a discrepancy between the training and inference phases. During training, each time-step is trained

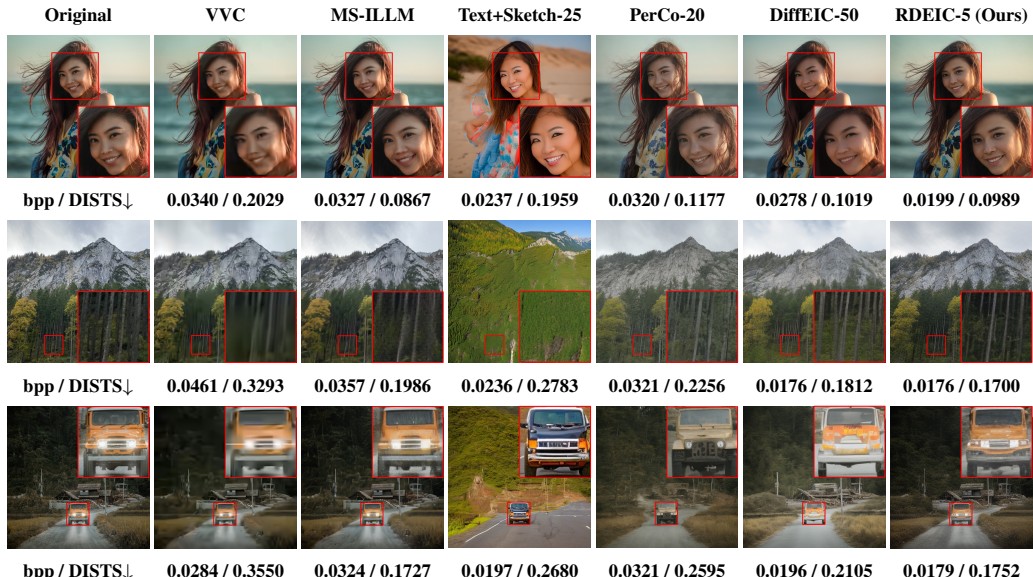

| Original | VVC | MS-ILLM | Text+Sketch-25 | PerCo-20 | DiffEIC-50 | RDEIC-5 (Ours) |
|---|---|---|---|---|---|---|
| bpp / DISTS↓ | 0.0340 / 0.2029 | 0.0327 / 0.0867 | 0.0237 / 0.1959 | 0.0320 / 0.1177 | 0.0278 / 0.1019 | 0.0199 / 0.0989 |
| bpp / DISTS↓ | 0.0461 / 0.3293 | 0.0357 / 0.1986 | 0.0236 / 0.2783 | 0.0321 / 0.2256 | 0.0176 / 0.1812 | 0.0176 / 0.1700 |
| bpp / DISTS↓ | 0.0284 / 0.3550 | 0.0324 / 0.1727 | 0.0197 / 0.2680 | 0.0321 / 0.2595 | 0.0196 / 0.2105 | 0.0179 / 0.1752 |

Figure 1: Qualitative comparison between the proposed RDEIC and state-of-the-art methods. The number of denoising steps is written after the name, e.g. DiffEIC-50 means 50 diffusion steps are used by DiffEIC. The bpp and DISTS of each method are shown at the bottom of each image.

independently, which is well-suited for image generation tasks where diversity (or randomness) is encouraged (Ho et al., 2020). However, this training approach is not optimal for image compression where consistency between the reconstruction and the raw image is crucial.

In this work, we propose **R**elay **R**esidual **D**iffusion **E**xtreme **I**mage **C**ompression (**RDEIC**) to overcome the three limitations mentioned above. To overcome the first two limitations, we proposed a novel relay residual diffusion framework. Specifically, we construct the starting point using the compressed latent features combined with slight noise, transitioning between the starting point and target latent features by shifting the residual between them. This approach significantly reduces the number of denoising steps required for reconstruction while ensures that the starting point retains most of the information from the compressed features, providing a strong foundation for subsequent detail generation. To leverage the robust generative capability of pre-trained stable diffusion for extreme image compression, we derive a novel residual diffusion equation directly from stable diffusion's diffusion equation, rather than designing a diffusion equation from scratch as Yue et al. (2023). To address the third limitation, we introduce a fixed-step fine-tuning strategy to eliminate the discrepancy between the training and inference phases. By fine-tuning RDEIC throughout the entire reconstruction process, we further improve the reconstruction quality. Moreover, to meet users' diverse requirements, we introduce a controllable detail generation method that achieves a trade-off between smoothness and sharpness by adjusting the intensity of high-frequency components in the reconstructions. As shown in Fig. 1, the proposed RDEIC achieves state-of-the-art perceptual performance at extremely low bitrates, and significantly outperforms existing diffusion-based extreme image compression methods with fewer inference steps.

In summary, our contributions are as follows:

- We propose RDEIC, a novel diffusion model for extreme image compression that outperforms existing diffusion-based extreme image compression methods in both reconstruction quality and efficiency.
- We propose a relay residual diffusion process that seamlessly integrates pre-trained stable diffusion. To the best of our knowledge, we are the first to successfully integrate stable diffusion into a residual diffusion framework.
- To eliminate the discrepancy between the training and inference phases, we design a fixed-step fine-tuning strategy that refines the model through the entire reconstruction process, further improving reconstruction quality.

- We introduce a controllable detail generation method to balance smoothness and sharpness, allowing users to explore and customize outputs according to their personal preferences.

## 2 RELATED WORK

**Learned Image Compression.** As a pioneer work, Ballé et al. (2017) proposed an end-to-end image compression framework to jointly optimize the rate-distortion performance. Ballé et al. (2018) later introduced a hyperprior to reduce spatial dependencies in the latent representation, greatly enhancing performance. Subsequent works further improved compression models by developing various nonlinear transforms (Xie et al., 2021; He et al., 2022; Liu et al., 2023; Li et al., 2024a) and entropy models (Minnen et al., 2018; Minnen & Singh, 2020; He et al., 2021; Qian et al., 2021). However, optimization for rate-distortion alone often results in unrealistic reconstructions at low bitrates, typically manifested as blurring or over-smoothness (Blau & Michaeli, 2019). To improve perceptual quality, generative models have been integrated into compression methods. Agustsson et al. (2019) added an adversarial loss for lost details generation. Mentzer et al. (2020) explored the generator and discriminator architectures, as well as training strategies for perceptual image compression. Muckley et al. (2023) introduced a local adversarial discriminator to enhance statistical fidelity. With the advancement of diffusion models, some efforts have been made to apply diffusion models to image compression. For instance, Yang & Mandt (2023) innovatively introduced a conditional diffusion model as decoder for image compression. Kuang et al. (2024) proposed a consistency guidance architecture to guide the diffusion model in stably reconstructing high-quality images.

**Extreme Image Compression.** In recent years, extreme image compression has garnered increasing attention, aiming to compress image to extremely low bitrates, often below 0.1 bpp, while maintaining visually acceptable image quality. Gao et al. (2023) leveraged the information-lossless property of invertible neural networks to mitigate the significant information loss in extreme image compression. Jiang et al. (2023) treated text descriptions as prior to ensure semantic consistency between the reconstructions and the raw images. Wei et al. (2024) achieved extreme image compression by rescaling images using extreme scaling factors. Lu et al. (2024) combined continuous and codebook-based discrete features to reconstruct high-quality images at extremely low bitrates. Inspired by the great success of T2I diffusion models in various image restoration tasks (Lin et al., 2023; Wang et al., 2024), some methods have incorporated T2I diffusion models into extreme image compression frameworks. Lei et al. (2023) utilized a pre-trained ControlNet (Zhang et al., 2023) to reconstruct images based on corresponding short text prompts and binary contour sketches. Careil et al. (2024) conditioned iterative diffusion models on vector-quantized latent image representations and textual image descriptions. Li et al. (2024b) combined compressive VAEs with pre-trained T2I diffusion models to achieve realistic reconstructions at extremely low bitrates. However, constrained by the inherent characteristics of diffusion models, these diffusion-based extreme image compression methods are limited in both fidelity and efficiency. In this paper, we propose a solution to these limitations through a relay residual diffusion framework and a fixed-step fine-tuning strategy.

**Relay Diffusion.** Conventional diffusion models, such as denoising diffusion probabilistic models (DDPM) (Ho et al., 2020) and its variants, have achieved remarkable results in low-resolution scenarios but face substantial challenges in terms of computational efficiency and performance when applied to higher resolutions. To overcome this, cascaded diffusion methods (Ho et al., 2022; Saharia et al., 2022) decompose the image generation into multiple stages, with each stage responsible for super-resolution conditioning on the previous one. However, these methods still require complete resampling at each stage, leading to inefficiencies and potential mismatches among different resolutions.

Relay diffusion, as proposed by Teng et al. (2024), extends the cascaded framework by continuing the diffusion process directly from the low-resolution output rather than restarting from pure noise, which allows the higher-resolution stages to correct artifacts from earlier stages. This design is particularly well-suited for tasks such as image restoration and image compression, where degraded images or features are available. PASD (Yang et al., 2023) and SeeSR (Wu et al., 2024) directly embed the LR latent into the initial random noise during the inference process to alleviate the inconsistency between training and inference. ResShift (Yue et al., 2023) further constructs a Markov chain that transfers between degraded and target features by shifting the residual between them, substantially improving the transition efficiency. However, its redesigned diffusion equation

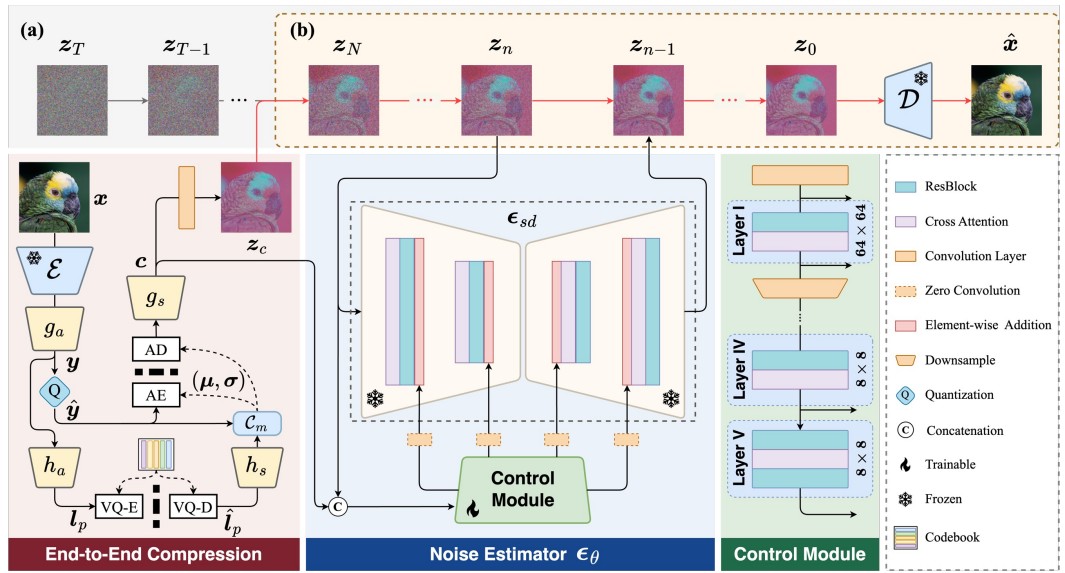

Figure 2: The proposed RDEIC. We first map a raw image $x$ into the latent space using the encoder $\mathcal{E}$ and then perform end-to-end lossy compression to get compressed latent features $z_c$. We then use $z_c$ with added noise as the starting point and apply a denoising process to reconstruct the noise-free latent feature $z_0$. The decoder $\mathcal{D}$ maps $z_0$ back to the pixel space, to get the reconstructed image $\hat{x}$. (a) Vanilla diffusion framework that starts from pure noise. (b) The proposed relay residual diffusion framework that starts from compressed latent features with added noise.

and noise schedule prevent it from leveraging the robust generative capability of pre-trained stable diffusion. In this work, we directly derive a new residual diffusion equation from stable diffusion's diffusion equation, enabling seamlessly integration of stable diffusion to leverage its robust generative capability.

## 3 METHODOLOGY

### 3.1 OVERALL FRAMEWORK

Fig. 2 shows an overview of the proposed RDEIC network. We first use an encoder $\mathcal{E}$ and analysis transform $g_a$ to convert the input image $x$ to its latent representation $y$. Then we perform hyper transform coding on $y$ with the categorical hyper model (Jia et al., 2024) and use the space-channel context model $\mathcal{C}_m$ to predict the entropy parameters $(\mu, \sigma)$ to estimate the distribution of quantized latent representation $\hat{y}$ (He et al., 2022). The side information $l_p$ is quantized through vector-quantization, i.e., $\hat{l}_p$ is the mapping of $l_p$ to its closest codebook entry. Subsequently, the synthesis transform $g_s$ is used to obtain the image content dependent features $z_c$. Random noise is then added to $z_c$, which is the starting point for reconstructing the noise-free latent features $z_0$ through an iterative denoising process. The denoising process is implemented by a frozen pre-trained noise estimator $\epsilon_{sd}$ of stable diffusion with trainable control network for intermediate feature modulation. Finally, the reconstructed image $\hat{x}$ is decoded from $z_0$ using the decoder $\mathcal{D}$.

### 3.2 ACCELERATING DENOISING PROCESS WITH RELAY RESIDUAL DIFFUSION

Following stable diffusion, existing diffusion-based extreme image compression methods obtain the noisy latent by adding Gaussian noise with variance $\beta_t \in (0, 1)$ to the noise-free latent features $z_0$:

$$z_t = \sqrt{\bar{\alpha}_t}z_0 + \sqrt{1-\bar{\alpha}_t}\epsilon_t, \ t = 1, 2, \cdots, T, \tag{1}$$

where $\epsilon_t \sim \mathcal{N}(0, I)$, $\alpha_t = 1 - \beta_t$ and $\bar{\alpha}_t = \prod_{i=1}^{t} \alpha_i$. When $t$ is large enough, the noisy latent $z_t$ is nearly a standard Gaussian distribution. In practice, $T$ is typically very large, e.g., 1000, and

pure noise is set as the starting point for the reverse diffusion process. However, this approach is not optimal for the image compression task, where the compressed latent features $z_c$ are available.

To this end, we set the starting point to $z_N = \sqrt{\bar{\alpha}_N}z_c + \sqrt{1-\bar{\alpha}_N}\epsilon_N$, where $N \ll T$. Our relay residual diffusion is thus defined as:

$$z_n = \sqrt{\bar{\alpha}_n}(z_0 + \eta_n e) + \sqrt{1-\bar{\alpha}_n}\epsilon_n, \ n = 1, 2, \cdots, N, \tag{2}$$

where $e$ denotes the residual between $z_c$ and $z_0$, i.e., $e = z_c - z_0$, and $\{\eta_n\}_{n=1}^N$ is a weight sequence that satisfies $\eta_1 \to 0$ and $\eta_N = 1$. Since the residual $e$ is unavailable during inference, we refer to DDIM (Song et al., 2021) and assume that $z_{n-1}$ is a linear combination of $z_n$ and $z_0$:

$$z_{n-1} = k_n z_0 + m_n z_n + \sigma_n \epsilon, \tag{3}$$

where we set $\sigma_n = 0$ for simplicity. Combining Eq. (2) and Eq. (3), we get

$$\frac{\eta_n}{\eta_{n-1}} = \frac{\sqrt{1-\bar{\alpha}_n}/\sqrt{\bar{\alpha}_n}}{\sqrt{1-\bar{\alpha}_{n-1}}/\sqrt{\bar{\alpha}_{n-1}}} \to \eta_n = \lambda \frac{\sqrt{1-\bar{\alpha}_n}}{\sqrt{\bar{\alpha}_n}}, \tag{4}$$

where we set $\lambda = \frac{\sqrt{\bar{\alpha}_N}}{\sqrt{1-\bar{\alpha}_N}}$ to ensure $\eta_N = 1$. Detailed derivation is presented in Appendix A. Substituting Eq. (4) into Eq. (2), the diffusion process can be further written as follows:

$$z_n = \sqrt{\bar{\alpha}_n}(z_0 + \lambda \frac{\sqrt{1-\bar{\alpha}_n}}{\sqrt{\bar{\alpha}_n}}e) + \sqrt{1-\bar{\alpha}_n}\epsilon_n, \tag{5}$$

$$= \sqrt{\bar{\alpha}_n}z_0 + \sqrt{1-\bar{\alpha}_n}\underbrace{(\lambda e + \epsilon_n)}_{\tilde{\epsilon}_n}. \tag{6}$$

Since Eq. (6) has the same structure as Eq. (1), we can easily incorporate stable diffusion into our framework. For the denoising process, the noise estimator $\epsilon_\theta$ is learned to predict $\tilde{\epsilon}_n$ at each time-step $n$. The optimization of noise estimator $\epsilon_\theta$ is defined as

$$\mathcal{L}_{ne} = \mathbb{E}_{z_0, z_c, c, n, \epsilon_n} \|z_0 - \hat{z}_0\|_2^2, \tag{7}$$

$$= \omega_n \mathbb{E}_{z_0, z_c, c, n, \epsilon_n} \|\tilde{\epsilon}_n - \epsilon_\theta(z_n, c, n)\|_2^2, \tag{8}$$

where $\omega_n = \frac{1-\bar{\alpha}_n}{\bar{\alpha}_n}$. After that, we can start from the compressed latent features $z_c$ and reconstruct the image using Eq. 3 without knowing the residual $e$.

### 3.3 FIXED-STEP FINE-TUNING STRATEGY

Most existing diffusion-based image compression methods adopt the same training strategy as DDPM (Ho et al., 2020), where each time-step is trained independently. However, the lack of co-ordination among time-steps can lead to error accumulation and suboptimal reconstruction quality. To address this issue, we employ a two-stage training strategy. As shown in Fig. 3(a), we first train each time-step $n$ independently, allowing the model to learn to remove noise and residuals at each step. The optimization objective consists of the rate-distortion loss, codebook loss (Van Den Oord et al., 2017) and noise estimation loss:

$$\mathcal{L}_{stage\ I} = \underbrace{\lambda_r \|z_0 - z_c\|_2^2 + R(\hat{y})}_{rate-distortion\ loss\ \mathcal{L}_{rd}} + \underbrace{\|sg(l_p) - \hat{l}_p\|_2^2 + \beta \|sg(\hat{l}_p) - l_p\|_2^2}_{codebook\ loss\ \mathcal{L}_{cb}} + \lambda_r \mathcal{L}_{ne}, \tag{9}$$

where $\lambda_r$ is the hyper-parameter that controls the trade-off, $R(\cdot)$ denotes the estimated rate, $sg(\cdot)$ denotes the stop-gradient operator, and $\beta = 0.25$. Thanks to the proposed relay residual diffusion framework, we can achieve high-quality reconstruction in fewer than 5 denoising steps, as demonstrated in Fig. 7. This efficiency allows us to fine-tune the model using the entire reconstruction process with limited computational resources.

To this end, we further employ a fixed-step fine-tuning strategy to eliminate the discrepancy between the training and inference phases. As shown in Fig. 3(b), in each training step, we utilize spaced DDPM sampling (Nichol & Dhariwal, 2021) with $L$ fixed time-steps to reconstruct the noise-free latent features $\hat{z}_0$ from the starting point $z_N$ and map $\hat{z}_0$ back to the pixel space $\hat{x} = \mathcal{D}(\hat{z}_0)$. The loss function used in this stage is as follows:

$$L_{stage\ II} = \mathcal{L}_{rd} + \mathcal{L}_{cb} + \lambda_r \|z_0 - \hat{z}_0\|_2^2 + \lambda_r(\|x - \hat{x}\|_2^2 + \lambda_{lpips}\mathcal{L}_{lpips}(x, \hat{x})), \tag{10}$$

where $\mathcal{L}_{lpips}$ denotes the LPIPS loss and $\lambda_{lpips} = 0.5$ is the weight of the LPIPS loss. By fine-tuning the model using the entire reconstruction process, we achieve significant performance improvement.

Figure 3: The two-stage training strategy of RDEIC. (a) Independent training: we randomly pick a time-step $n$ and train each time-step $n$ independently. This ensures that the model effectively learns to remove added noise and residuals at every step. (b) Fixed-step fine-tuning: $L$ fixed denoising steps are used to iteratively reconstruct a noise-free latent features $\hat{z}_0$ from $z_N$, which is consistent with the inference phase.

### 3.4 Controllable Detail Generation

Although the fixed-step fine-tuning strategy significantly improves reconstruction quality, it requires a fixed number of denoising steps in the inference phase, making it impossible to achieve a trade-off between smoothness and sharpness by adjusting the number of denoising steps (Li et al., 2024b). To address this limitation, we introduce a controllable detail generation method that allows us to dynamically balance smoothness and sharpness without being constrained by the fixed-step requirement, which enables more versatile and user-specific image reconstructions.

Since the compressed latent feature already contains image information, directly using stable diffusion's noise estimator $\epsilon_{sd}$ to predict noise $\epsilon_{sd}(z_n, n)$ results in low-frequency reconstructed images, as shown in the second column of Fig. 8 and Fig. 17. Inspired by classifier-free guidance (Ho & Salimans, 2021), we decompose the predicted noise $\epsilon_\theta(z_n, c, n)$ into a low-frequency control component $\epsilon_{sd}(z_n, n)$ and a high-frequency control component $\epsilon_\theta(z_n, c, n) - \epsilon_{sd}(z_n, n)$, and control the balance between smoothness and sharpness by adjusting the intensity of the high-frequency control component:

$$\hat{\epsilon}_n = \epsilon_{sd}(z_n, n) + \lambda_s(\epsilon_\theta(z_n, c, n) - \epsilon_{sd}(z_n, n)), \tag{11}$$

where $\lambda_s$ is the guidance scale. By adjusting the value of $\lambda_s$, we can regulate the amount of high-frequency details introduced into the reconstructed image. In the experiments, we set $\lambda_s = 1$ by default unless otherwise specified.

## 4 Experiments

### 4.1 Experimental Setup

**Datasets.** The proposed RDEIC is trained on the **LSDIR** (Li et al., 2023) dataset, which contains 84,911 high-quality images. For evaluation, we use three common benchmark datasets, i.e., the **Kodak** (Franzen, 1999) dataset with 24 natural images of $768 \times 512$ pixels, the **Tecnick** (Asuni & Giachetti, 2014) dataset with 140 images of $1200 \times 1200$ pixels, and the **CLIC2020** (Toderici et al., 2020) dataset with 428 high-quality images. For the Tecnick and CLIC2020 datasets, we resize the images so that the shorter dimension is equal to 768 and then center-crop them with $768 \times 768$ spatial resolution (Yang & Mandt, 2023).

**Implementation details.** We use Stable Diffusion 2.1-base[1] as the specific implementation of stable diffusion. Throughout all our experiments, the weights of stable diffusion remain frozen. To achieve different compression ratios, we train five models with $\lambda_r$ selected from $\{2, 1, 0.5, 0.25, 0.1\}$. The total number $N$ of denoising steps is set to 300. The size of codebook is set to 16384. For the fixed-step fine-tuning strategy, we use varying numbers of denoising steps to fine-tune models

[1]https://huggingface.co/stabilityai/stable-diffusion-2-1-base

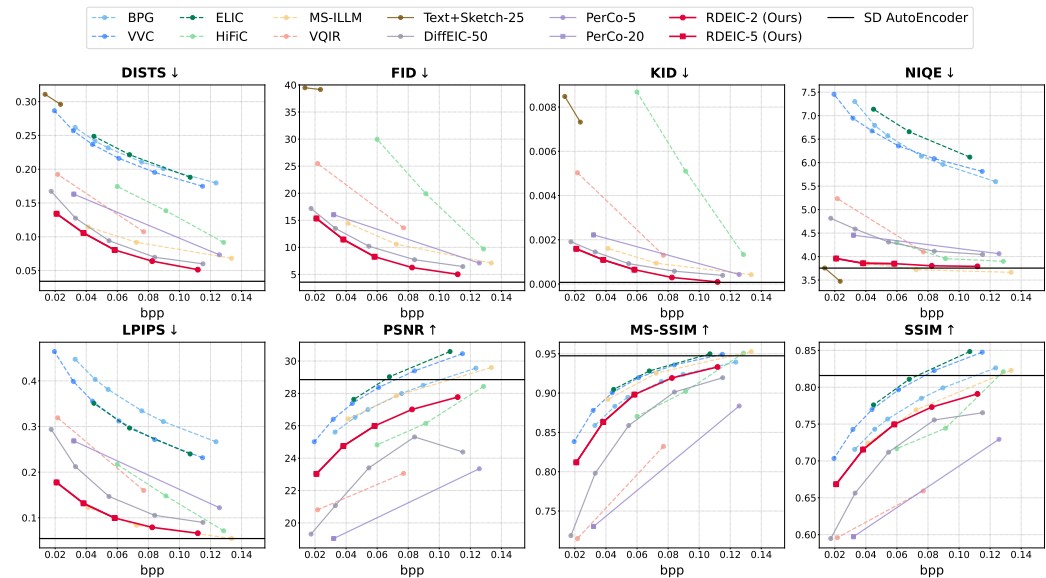

Figure 4: Quantitative comparisons with state-of-the-art methods on the CLIC2020 dataset. Solid lines are used for diffusion-based methods, while dashed lines represent other methods. For RDEIC, we use 2 denoising steps for the two models with larger bpp and 5 steps for the remaining models.

with different compression ratios. Specifically, when $\lambda_r \in \{2, 1\}$, the fixed number $L$ is set to 2, otherwise, it is 5. All experiments are conducted on a single NVIDIA GeForce RTX 4090 GPU.

**Metrics.** For quantitative evaluation, we employ several established metrics to measure the visual quality of the reconstructed images, including reference perceptual metrics **LPIPS** (Zhang et al., 2018), **DISTS** (Ding et al., 2020), **FID** (Heusel et al., 2017) and **KID** (Bińkowski et al., 2018) and no-reference perceptual metric **NIQE** (Mittal et al., 2012). We also employ distortion metrics **PSNR**, **SSIM** and **MS-SSIM** (Wang et al., 2003) to measure the fidelity of reconstructions. Note that FID and KID are calculated on 256×256 patches according to Mentzer et al. (2020).

**Comparison methods.** We compare the proposed RDEIC with several representative extreme image compression methods, including the traditional standards: BPG (Bellard, 2014) and VVC (Bross et al., 2021); VAE-based method: ELIC (He et al., 2022); GANs-based methods: HiFiC (Mentzer et al., 2020), MS-ILLM (Muckley et al., 2023), and VQIR (Wei et al., 2024); and diffusion-based methods: Text+Sketch (Lei et al., 2023), PerCo (Careil et al., 2024),and DiffEIC (Li et al., 2024b). More details can be found in Appendix B.

## 4.2 EXPERIMENTAL RESULTS

**Quantitative comparisons.** Fig. 4 shows the performance of the proposed and compared methods on the CLIC2020 dataset. It can be observed that the proposed RDEIC demonstrates superior performance across different perceptual metrics compared to other methods, particularly achieving optimal results in DISTS, FID, and KID. For the distortion metrics, RDEIC significantly outperforms other diffusion-based methods, underscoring its superiority in maintaining consistency. Moreover, we report the performance of the SD autoencoder in Fig. 4 (see the black horizontal line), which represents the upper bound of RDEIC's performance. Compared to DiffEIC (Li et al., 2024b), which is also based on stable diffusion, RDEIC is significantly closer to this performance upper limit. To provide a more intuitive comparison of overall performance, we compute the BD-rate (Bjontegaard, 2001) for each metric. The results are shown in Table 3. The comparison results on the Tecnick and Kodak datasets are shown in Fig. 14 and Fig. 15, respectively.

**Qualitative comparisons.** Fig. 1 and Fig. 5 provides visual comparisons among the evaluated methods at extremely low bitrates. VVC (Bross et al., 2021) and MS-ILLM (Muckley et al., 2023) excel at reconstructing structural information, such as text, but falls significantly short in preserving

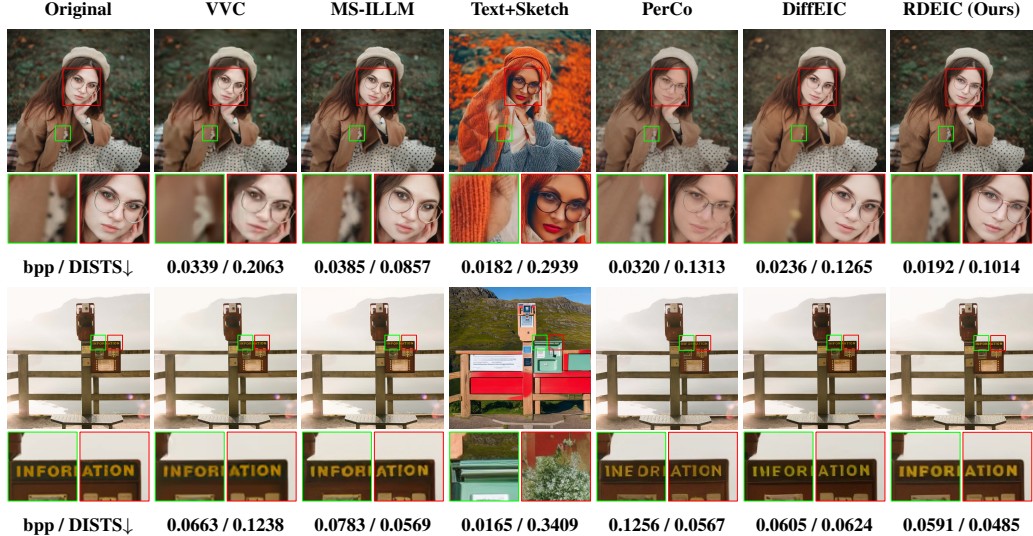

| | Original | VVC | MS-ILLM | Text+Sketch | PerCo | DiffEIC | RDEIC (Ours) |
|---|---|---|---|---|---|---|---|
| bpp / DISTS↓ | | 0.0339 / 0.2063 | 0.0385 / 0.0857 | 0.0182 / 0.2939 | 0.0320 / 0.1313 | 0.0236 / 0.1265 | 0.0192 / 0.1014 |
| bpp / DISTS↓ | | 0.0663 / 0.1238 | 0.0783 / 0.0569 | 0.0165 / 0.3409 | 0.1256 / 0.0567 | 0.0605 / 0.0624 | 0.0591 / 0.0485 |

Figure 5: Visual comparisons of our method to baselines on the CLIC2020 dataset. Compared to other methods, our method produces more realistic and faithful reconstructions.

Table 1: Encoding and decoding time (in seconds) on Kodak dataset. Decoding time is divided into the time spent in the denoising stage and the time spent in the remaining parts. DS denotes the number of denoising steps. The testing platform is RTX4090.

| Types | Methods | DS | Encoding Time | Decoding time | |
|---|---|---|---|---|---|
| | | | | Denoising Time | Remaining Time |
| VAE-based | ELIC | – | $0.056 \pm 0.006$ | – | $0.081 \pm 0.011$ |
| GAN-based | HiFiC | – | $0.038 \pm 0.004$ | – | $0.059 \pm 0.004$ |
| | MS-ILLM | – | $0.038 \pm 0.004$ | – | $0.059 \pm 0.004$ |
| | VQIR | – | $0.050 \pm 0.003$ | – | $0.179 \pm 0.005$ |
| Diffusion-based | Text+Sketch | 25 | $62.045 \pm 0.516$ | $8.483 \pm 0.344$ | $4.030 \pm 0.469$ |
| | DiffEIC | 50 | $0.128 \pm 0.005$ | $4.342 \pm 0.013$ | $0.228 \pm 0.026$ |
| | PerCo | 5 | $0.236 \pm 0.040$ | $0.623 \pm 0.003$ | $0.186 \pm 0.002$ |
| | | 20 | $0.236 \pm 0.040$ | $2.495 \pm 0.009$ | $0.186 \pm 0.002$ |
| | RDEIC (Ours) | 2 | $0.119 \pm 0.003$ | $0.173 \pm 0.001$ | $0.198 \pm 0.003$ |
| | | 5 | $0.119 \pm 0.003$ | $0.434 \pm 0.002$ | $0.198 \pm 0.003$ |

textures and fine details. Diffusion-based Text+Sketch (Lei et al., 2023), PerCo (Careil et al., 2024) and DiffEIC (Li et al., 2024b) achieve realistic reconstruction at extremely low bitrates but often generate details and structures that are inconsistent with the original image. In comparison, the proposed RDEIC produces reconstructions with higher visual quality, fewer artifacts, and more faithful details.

**Complexity comparisons.** Table 1 summarizes the average encoding/decoding times along with standard deviations for different methods on the Kodak dataset. For diffusion-based methods, decoding time is divided into denoising time and remaining time. Due to relay on stable diffusion, diffusion-based extreme image compression methods have higher encoding and decoding complexity than other learned-based methods. By reducing the number of denoising steps required for reconstruction, the denoising time of RDEIC is significantly lower than that of other diffusion-based methods. For instance, compared to DiffEIC (Li et al., 2024b), our RDEIC is approximately 10× to 25× faster in terms of denoising time.

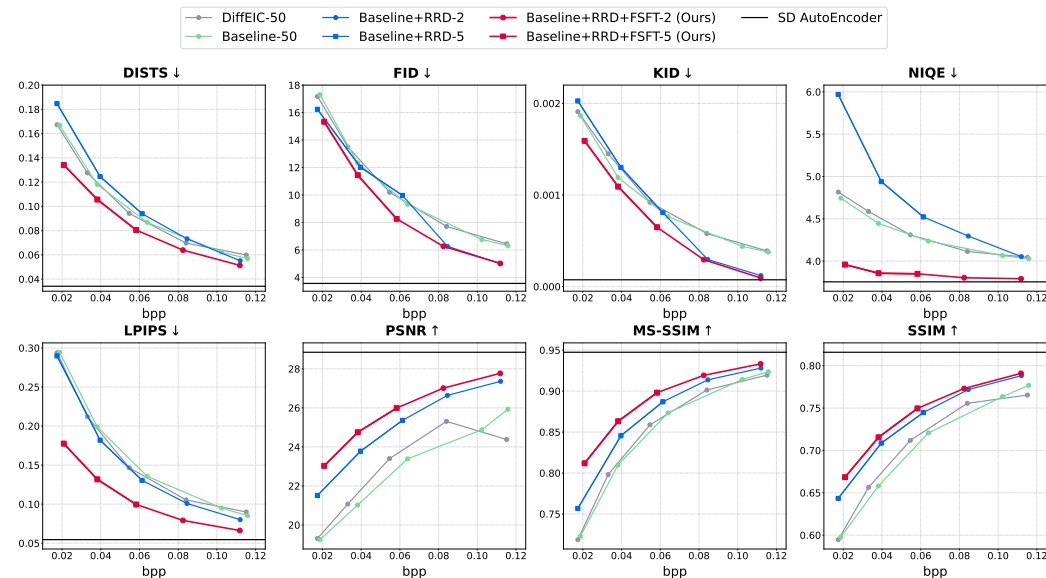

Figure 6: Ablation studies on the proposed relay residual diffusion and fixed-step fine-tuning.

Table 2: The impact of RRD and FSFT on performance (left) and speed (right). Performance is represented by BD-rate (%), using DiffEIC-50 as the anchor. Distortion metrics include PSNR, MS-SSIM, and SSIM. Perceptual metrics include DISTS, FID, KID, NIQE, and LPIPS. DS denotes the number of denoising steps. 2/5 denotes that we use 2 denoising steps for the two models with larger bpp and 5 steps for the remaining models. FSFT is a fine-tuning strategy that does not affect speed.

| Methods | DS | Distortion | Perception | Average |
|---|---|---|---|---|
| Baseline | 50 | 7.4 | -1.8 | 2.8 |
| +RRD | 2/5 | -31.0 | 12.7 | -9.1 |
| +RRD+FSFT | 2/5 | -42.2 | -36.6 | -39.4 |

| Methods | DS | Denoising Time | Speedup |
|---|---|---|---|
| Baseline | 50 | $4.349 \pm 0.013$ | $1\times$ |
| +RRD | 5 | $0.434 \pm 0.002$ | $10\times$ |
|  | 2 | $0.173 \pm 0.001$ | $25\times$ |

## 4.3 ABLATIONS

To provide a more comprehensive analysis of the proposed method, we conduct ablation studies, with the results presented in Fig. 6 and Table 2. For the baseline, we employ the same diffusion framework as DiffEIC (Li et al., 2024b), where the denoising process starts from pure noise. As shown in Fig. 6, our baseline performs similarly to DiffEIC (Li et al., 2024b).

**Effectiveness of relay residual diffusion.** We first investigate the effectiveness of our proposed relay residual diffusion framework. As shown in Fig. 6 and Table 2(left), by incorporating the proposed relay residual diffusion framework, we achieve better distortion performance and comparable perceptual performance with 2/5 denoising steps compared to the Baseline, which uses 50 denoising steps. The reason behind this is that starting from the compressed latent feature, instead of pure noise, avoids the error accumulation in the initial stage of the denoising process and provides a solid foundation for subsequent detail generation. Since the time required for the denoising stage is directly proportional to the number of denoising steps, incorporating RRD reduces the denoising time by a factor of $10\times$ to $25\times$ compared to the baseline, as shown in Table 2(right).

**Analyze of denoising steps.** Next, we analyze the impact of denoising steps on "Baseline+RRD" to select an appropriate value of $L$ for FSFT strategy. As shown in Fig. 7, for $\lambda_r \in \{2, 1\}$, the number of denoising steps has minimal effect on compression performance, so that we set $L$ to 2 in this case. For $\lambda_r \in \{0.5, 0.25, 0.1\}$, increasing the denoising steps achieves better perceptual results (lower LPIPS and DISTS values), but leads to degraded fidelity (lower PSNR and MS-SSIM values). To achieve a balance between fidelity and perceptual quality, we set $L$ to 5 here.

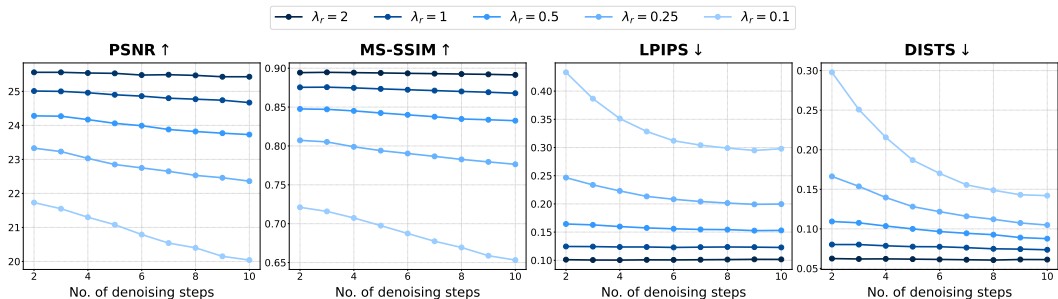

Figure 7: The impact of denoising steps on "Baseline+RRD".

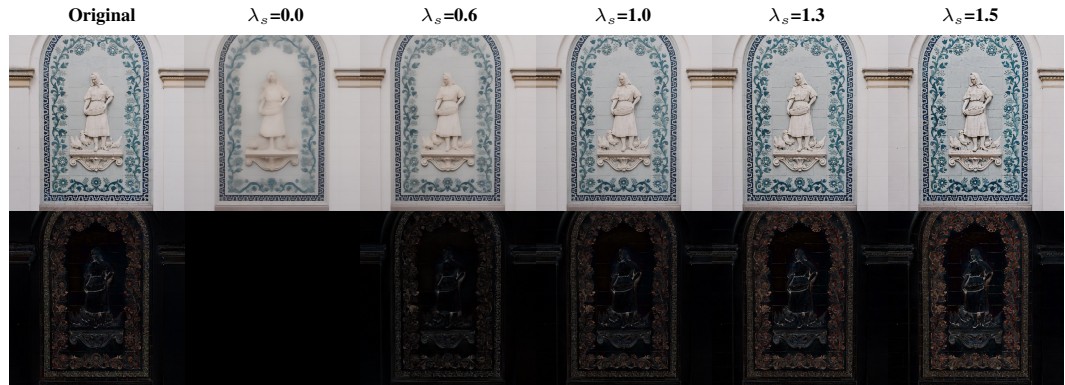

Figure 8: Balancing smoothness versus sharpness. The second row shows the absolute difference between the reconstructed images and the baseline ($\lambda_s = 0$).

**Effectiveness of fixed-step fine-tuning.** We further demonstrate the effectiveness of the FSFT strategy. As shown in Fig. 6 and Table 2(left), the FSFT strategy significantly improves reconstruction performance across all metrics, indicating that it effectively eliminates the discrepancy between the training and inference phases. Furthermore, as FSFT is a fine-tuning strategy, it does not introduce any additional computational overhead during inference.

**Smoothness-sharpness trade-off.** To fully leverage the generative potential of pre-trained stable diffusion, we introduce a controllable detail generation method that allows users to explore and customize outputs according to their personal preferences. For this experiment, we used the model trained with $\lambda_r = 1$. The visualization result is shown in Fig. 8. We control the balance between smoothness and sharpness by adjusting the parameter $\lambda_s$, which regulates the amount of high-frequency details introduced into the reconstructed image. Specifically, as the value of $\lambda_s$ increases, the image transitions from a smooth appearance to a progressively sharper and more detailed reconstruction. Additional results are provided in Fig. 17, Fig. 18, and Fig. 19 in Appendix D.

## 5 CONCLUSION

In this paper, we propose an innovative relay residual diffusion-based method (RDEIC) for extreme image compression. Unlike most existing diffusion-based methods that start from pure noise, RDEIC takes the compressed latent features of the input image with added noise as the starting point and reconstructs the image by iteratively removing the noise and reducing the residual between the compressed latent features and the target latent features. Extensive experiments have demonstrated the superior performance of our RDEIC over existing state-of-the-art methods in terms of both reconstruction quality and computational complexity.

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

# A  MATHEMATICAL DETAILS

**Derivation of Eq. (4).** First, according to Eq. (2), $z_{n-1}$ can be sampled as:

$$z_{n-1} = \sqrt{\bar{\alpha}_{n-1}}(z_0 + \eta_{n-1}e) + \sqrt{1 - \bar{\alpha}_{n-1}}\epsilon_{n-1}, \tag{12}$$

$$= \sqrt{\bar{\alpha}_{n-1}}z_0 + \sqrt{\bar{\alpha}_{n-1}}\eta_{n-1}e + \underbrace{\sqrt{1 - \bar{\alpha}_{n-1}}\epsilon_{n-1}}_{\sim \mathcal{N}(0,(1-\bar{\alpha}_{n-1})I)}, \tag{13}$$

where $\epsilon_{n-1} \sim \mathcal{N}(0, I)$. Second, for $z_n$ defined in Eq. (2) and $z_{n-1}$ defined in Eq. (3), we have:

$$z_{n-1} = k_n z_0 + m_n z_n + \sigma_n \epsilon, \tag{14}$$

$$= k_n z_0 + m_n(\sqrt{\bar{\alpha}_n}(z_0 + \eta_n e) + \sqrt{1 - \bar{\alpha}_n}\epsilon_n) + \sigma_n \epsilon, \tag{15}$$

$$= (k_n + m_n\sqrt{\bar{\alpha}_n})z_0 + m_n\sqrt{\bar{\alpha}_n}\eta_n e + \underbrace{m_n\sqrt{1 - \bar{\alpha}_n}\epsilon_n + \sigma_n\epsilon}_{\sim \mathcal{N}(0,(m_n^2(1-\bar{\alpha}_n)+\sigma_n^2)I)}, \tag{16}$$

where $\epsilon_n \sim \mathcal{N}(0, I)$ and $\epsilon \sim \mathcal{N}(0, I)$. By combining Eq. (13) and Eq. (16), we obtain the following equations:

$$\begin{cases} \sqrt{\bar{\alpha}_{n-1}} = k_n + m_n\sqrt{\bar{\alpha}_n}, \\ \sqrt{\bar{\alpha}_{n-1}}\eta_{n-1} = m_n\sqrt{\bar{\alpha}_n}\eta_n, \\ 1 - \bar{\alpha}_{n-1} = m_n^2(1 - \bar{\alpha}_n) + \sigma_n^2. \end{cases} \tag{17}$$

Note that, referring to DDIM (Song et al., 2021), we set $\sigma_n = 0$ for simplicity. By solving Eq. (17), we have:

$$k_n = \sqrt{\bar{\alpha}_{n-1}} - \sqrt{\frac{1 - \bar{\alpha}_{n-1}}{1 - \bar{\alpha}_n}}\sqrt{\bar{\alpha}_n}, \ m_n = \sqrt{\frac{1 - \bar{\alpha}_{n-1}}{1 - \bar{\alpha}_n}}, \ \frac{\eta_n}{\eta_{n-1}} = \frac{\sqrt{1 - \bar{\alpha}_n}/\sqrt{\bar{\alpha}_n}}{\sqrt{1 - \bar{\alpha}_{n-1}}/\sqrt{\bar{\alpha}_{n-1}}}. \tag{18}$$

Therefore, $\eta_n$ can be defined as:

$$\eta_n = \lambda \frac{\sqrt{1 - \bar{\alpha}_n}}{\sqrt{\bar{\alpha}_n}}, \tag{19}$$

where we set $\lambda = \frac{\sqrt{\bar{\alpha}_N}}{\sqrt{1 - \bar{\alpha}_N}}$ to ensure $\eta_N = 1$.

**Derivation of Eq. (8).** Substituting Eq. (6) into Eq. (7), we have:

$$\|z_0 - \hat{z}_0\|_2^2 = \|(\frac{z_n}{\sqrt{\bar{\alpha}_n}} - \frac{\sqrt{1 - \bar{\alpha}_n}}{\sqrt{\bar{\alpha}_n}}\tilde{\epsilon}_n) - (\frac{z_n}{\sqrt{\bar{\alpha}_n}} - \frac{\sqrt{1 - \bar{\alpha}_n}}{\sqrt{\bar{\alpha}_n}}\epsilon_\theta(z_n, c, n))\|_2^2, \tag{20}$$

$$= \|\frac{\sqrt{1 - \bar{\alpha}_n}}{\sqrt{\bar{\alpha}_n}}\tilde{\epsilon}_n - \frac{\sqrt{1 - \bar{\alpha}_n}}{\sqrt{\bar{\alpha}_n}}\epsilon_\theta(z_n, c, n)\|_2^2, \tag{21}$$

$$= \frac{1 - \bar{\alpha}_n}{\bar{\alpha}_n}\|\tilde{\epsilon}_n - \epsilon_\theta(z_n, c, n)\|_2^2. \tag{22}$$

# B  EXPERIMENTAL DETAILS

**Evaluation of third-party models.** The quality factor of BPG (Bellard, 2014) was selected from $\{43, 45, 46, 48, 49, 51\}$. For VVC (Bross et al., 2021), we used the reference software VTM-23.0[2] with intra configuration. The quality factor was selected from the set $\{41, 43, 45, 47, 49, 52\}$. To compare ELIC (He et al., 2022) and HiFiC (Mentzer et al., 2020) at extremely low bitrates, we utilized their PyTorch implementation[34] and retrained the model to achieve higher compression ratios, enabling a more direct comparison with our proposed method. For PerCo (Careil et al., 2024), since the official source codes and models are not available, we used a reproduced version[5] as a substitute, which employs stable diffusion as the latent diffusion model. For MS-ILLM (Muckley et al., 2023), VQIR (Wei et al., 2024), Text+Sketch (Lei et al., 2023) and DiffEIC (Li et al., 2024b), we used the

---

[2] https://vcgit.hhi.fraunhofer.de/jvet/VVCSoftware_VTM/-/tree/VTM-23.0

[3] https://github.com/JiangWeibeta/ELIC

[4] https://github.com/Justin-Tan/high-fidelity-generative-compression

[5] https://github.com/Nikolai10/PerCo/tree/master

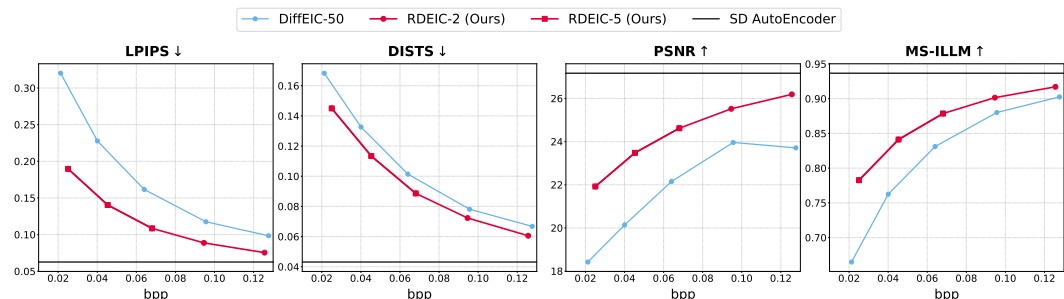

Figure 9: Quantitative performance on the MS-COCO 30k dataset.

publicly released checkpoints from their GitHub repositories, and used them for evaluation with the provided code.

**Additional implementation details.** We use Stable Diffusion 2.1-base as the specific implementation of stable diffusion. Throughout all our experiments, the weights of stable diffusion remain frozen. Similar to DiffEIC (Li et al., 2024b), the control module in our RDEIC has the same encoder and middle block architecture as stable diffusion and reduces the channel number to 20% of the original. The variance sequence $\{\beta_t\}_{t=1}^{T}$ used for adding noise is identical to that in Stable Diffusion. The number $N$ of denoising steps is set to 300. For the update of codebook, we use the clustering strategy proposed in CVQ-VAE (Zheng & Vedaldi, 2023).

For training, we use the Adam (Kingma & Ba, 2014) optimizer with $\beta_1 = 0.9$ and $\beta_2 = 0.999$ for a total of 300K iterations. To achieve different compression ratios, we train five models with $\lambda_r$ selected from $\{2, 1, 0.5, 0.25, 0.1\}$. The batch size is set to 4. As described in Section 3.3, the training process is divided into two stages. *1) Independent training.* During this stage, the initial learning rate is set to $1 \times 10^{-4}$ and images are randomly cropped to $512 \times 512$ patches. We first train the proposed RDEIC with $\lambda_r = 2$ for 100K iterations. The learning rate is then reduced to $2 \times 10^{-5}$ and the model is trained with target $\lambda_r$ for another 100K iterations. *2) Fixed-step fine-tuning.* In this stage, the learning rate is set to $2 \times 10^{-5}$ and images are randomly cropped to $256 \times 256$ patches. We fine-tune the model through the entire reconstruction process for 100K iterations. When $\lambda_r \in \{2, 1\}$, the fixed number $L$ is set to 2, otherwise, it is 5. All experiments are conducted on a single NVIDIA GeForce RTX 4090 GPU.

## C  FURTHER ABLATION EXPERIMENTS

**Robustness and generalization ability.** To assess the robustness and generalization ability of RDEIC, we conducted additional experiments on the larger MS-COCO 30k dataset, which comprises 30,000 images spanning a diverse range of categories and content types. This dataset was constructed by selecting the same images from the COCO2017 training set (Caesar et al., 2018) as Careil et al. (2024).

As shown in Fig. 9, RDEIC maintains consistent performance across this expanded dataset, demonstrating its ability to generalize effectively to unseen data, even in scenarios with more diverse and challenging content. Visualized examples of reconstructed images are provided in Fig. 16 to further illustrate the robustness of our approach.

**Role of the diffusion mechanism.** To further investigate the role of the diffusion mechanism in RDEIC, we design two variants for comparison: 1) **W/o denoising process**: In this variant, the compression module is trained jointly with the noise estimator, but the denoising process is bypassed during the inference phase. 2) **W/o diffusion mechanism**: In this variant, the compression module is trained independently, completely excluding the influence of the diffusion mechanism.

As shown in Fig. 10, bypassing the denoising process results in significant degradation, particularly in perceptual quality. This demonstrates that the diffusion mechanism plays a crucial role in enhancing perceptual quality during reconstruction. As shown in Fig. 11, the diffusion mechanism effectively adds realistic and visually pleasing details.

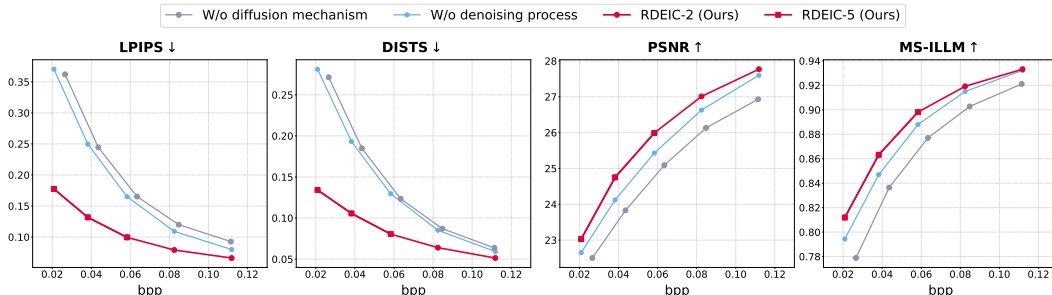

Figure 10: Ablation studies of the diffusion mechanism on CLIC2020 dataset. In the **W/o denoising process** setting, we train the compression module jointly with the noise estimator but bypass the denoising process during inference. In the **W/o diffusion mechanism** setting, we train the compression module independently, completely excluding the influence of the diffusion mechanism.

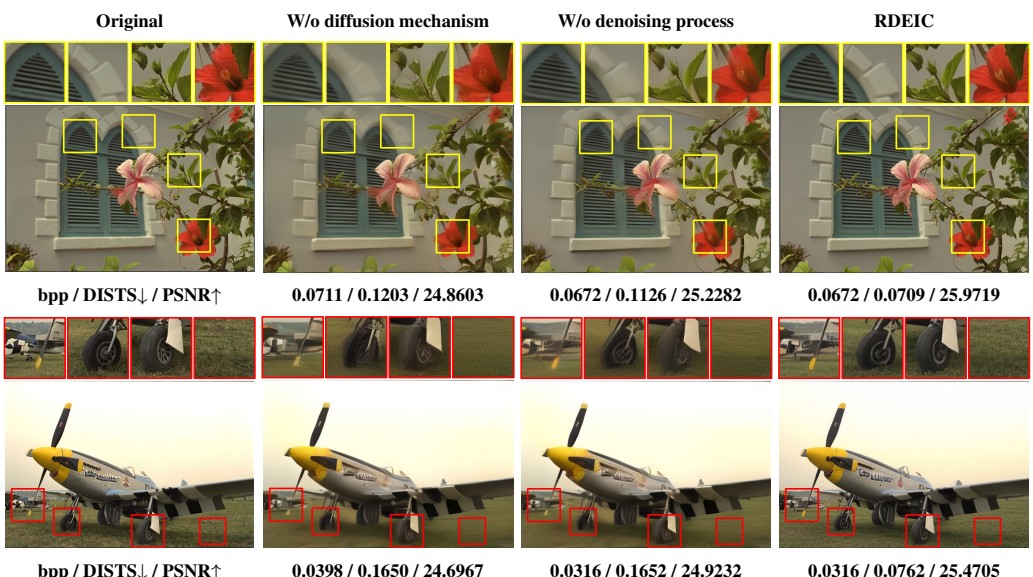

Figure 11: Impact of diffusion mechanism on reconstruction results.

By comparing the performance of **W/o diffusion mechanism** and **W/o denoising process** in Fig. 10 and Fig. 11, we observe that the compression module trained jointly with the noise estimator outperforms the one trained independently. This demonstrates that the diffusion mechanism also contributes to the compression module. Moreover, Fig. 12(a) visualizes an example of bit allocation. It is evident that the model trained jointly with the noise estimator allocates bits more efficiently, assigning fewer bits to flat regions (e.g., the sky in the image). Fig. 12(b) visualizes the cross-correlation between each spatial pixel in $(\boldsymbol{y} - \boldsymbol{\mu})/\boldsymbol{\sigma}$ and its surrounding positions. Specifically, the value at position $(i, j)$ represents cross-correlation between spatial locations $(x, y)$ and $(x+i, y+j)$ along the channel dimension, averaged across all images on Kodak dataset. It is evident that the model trained jointly with the noise estimator exhibits lower latent correlation, suggesting reduced redundancy and more compact feature representations. These results indicate that the diffusion mechanism provides additional guidance for optimizing the compression module during training, enabling it to learn more efficient and compact feature representations.

# D  ADDITIONAL EXPERIMENTAL RESULTS

**BD-rate (%) on the CLIC2020 dataset.** To provide a more intuitive comparison of overall performance on CLIC2020 dataset, we set DiffEIC (Li et al., 2024b) as the anchor and compute the

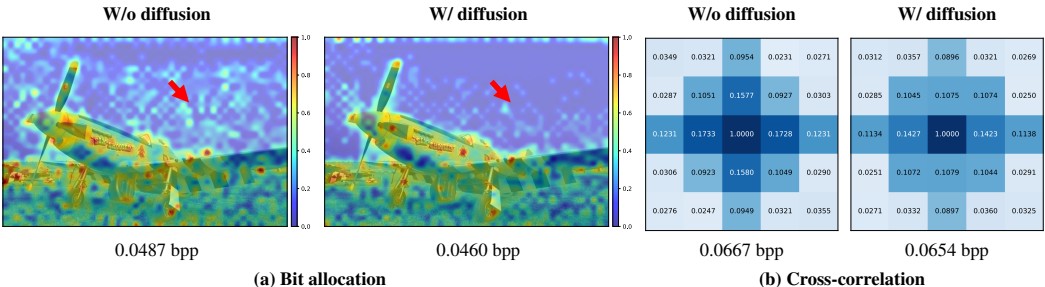

| W/o diffusion | W/ diffusion | W/o diffusion | W/ diffusion |
| --- | --- | --- | --- |
| 0.0487 bpp | 0.0460 bpp | 0.0667 bpp | 0.0654 bpp |

**(a) Bit allocation**      **(b) Cross-correlation**

Figure 12: Impact of the diffusion mechanism on the compression module. **W/o diffusion** denotes the compression module trained independently, while **W/ diffusion** denotes the compression module trained jointly with the noise estimator. All results are obtained from models trained with $\lambda_r = 0.5$. (a) An example of bit allocation on the Kodak dataset, with the values normalized for consistency. (b) Latent correlation of $(\boldsymbol{y} - \boldsymbol{\mu})/\boldsymbol{\sigma}$.

Table 3: BD-rate (%) for different methods on the CLIC2020 dataset with DiffEIC as the anchor. For distortion-oriented methods (i.e., BPG, VVC, and ELIC), we omit their perceptual metrics. The best and second best results are highlighted in **bold** and underline.

| Methods | Perception | | | | | Distortion | | | Average |
| --- | --- | --- | --- | --- | --- | --- | --- | --- | --- |
| | DISTS | FID | KID | NIQE | LPIPS | PSNR | MS-SSIM | SSIM | |
| BPG | – | – | – | – | – | -66.2 | -32.8 | -40.3 | – |
| VVC | – | – | – | – | – | -77.8 | -51.3 | -58.6 | – |
| ELIC | – | – | – | – | – | **-82.7** | **-54.6** | **-66.7** | – |
| HiFiC | 201.8 | 248.2 | 372.6 | -28.7 | 63.4 | -29.1 | 2.7 | 14.7 | 105.7 |
| VQIR | 71.8 | 183.9 | 156.7 | 32.4 | 51.3 | 16.4 | 43.9 | 57.8 | 76.8 |
| PerCo | 66.1 | 67.6 | 65.1 | 5.2 | 67.7 | 33.9 | 69.2 | 77.7 | 56.6 |
| MS-ILLM | 28.5 | 40.9 | 34.6 | **-85.4** | **-44.7** | -75.4 | -44.7 | -38.5 | -21.5 |
| RDEIC(Ours) | **-17.9** | **-18.3** | **-22.1** | -83.7 | -40.8 | -61.3 | -32.7 | -32.7 | **-38.7** |

BD-rate (Bjontegaard, 2001) for each metric. As shown in Table 3, our method outperforms all perception-oriented comparison methods, achieving the lowest average BD-rate value among them.

**Quantitative comparisons on the Tecnick and Kodak datasets.** We present the performance of the proposed and compared methods on the Tecnick and Kodak datasets in Fig. 14 and Fig. 15, respectively. The proposed RDEIC achieves state-of-the-art perceptual performance and significantly outperforms other diffusion-based methods in terms of distortion metrics. Since the Kodak dataset is too small to reliably calculate FID and KID scores, we do not report these results for this dataset.

**Smoothness-sharpness trade-off.** As shown in Fig. 17, Fig. 18, and Fig. 19, we control the balance between smoothness and sharpness by adjusting the parameter $\lambda_s$, which regulates the amount of high-frequency details introduced into the reconstructed image.

## E  LIMITATIONS

Using pre-trained stable diffusion may generate hallucinated lower-level details at extremely low bitrates. For instance, as shown in Fig. 13, the generated human faces appear realistic but are inaccurate, which may lead to a misrepresentation of the person's identity. Furthermore, although the proposed RDEIC has shown promising compression results, the potential of incorporating a text-driven strategy has not yet been explored within our framework. We leave detailed study of this to future work.

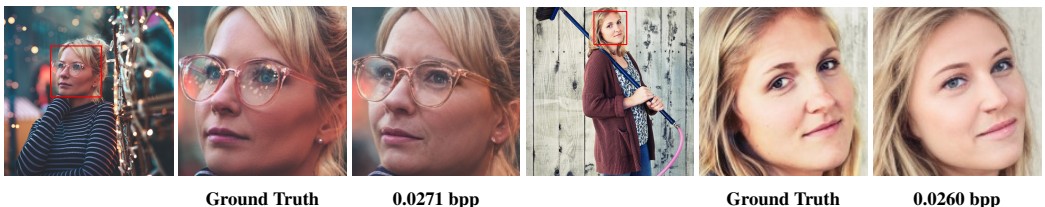

Figure 13: Faces generated at extremely low bitrates.

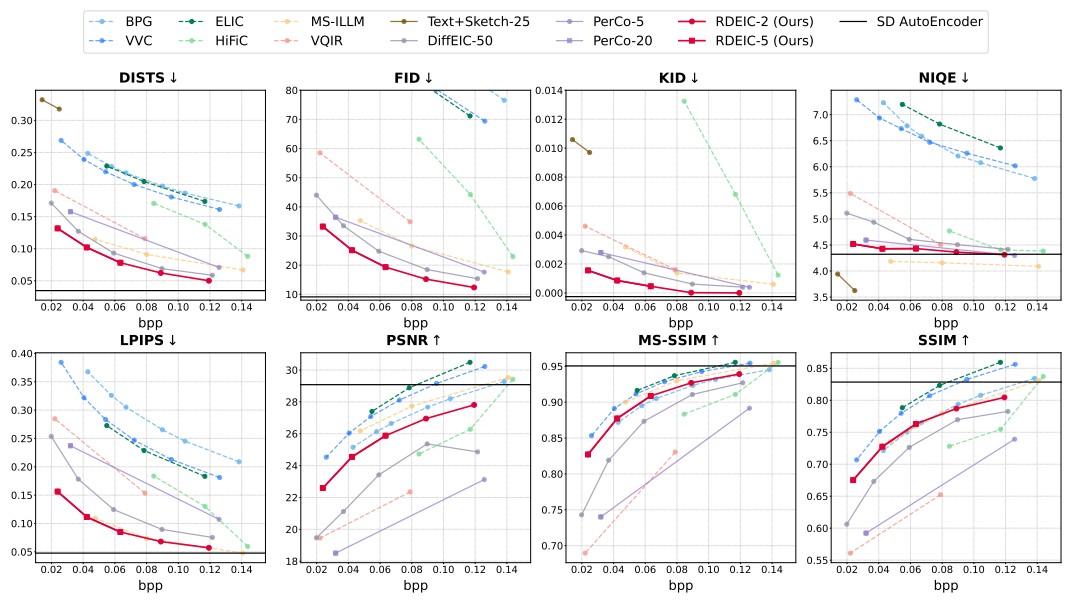

Figure 14: Quantitative comparisons with state-of-the-art methods on the Tecnick dataset.

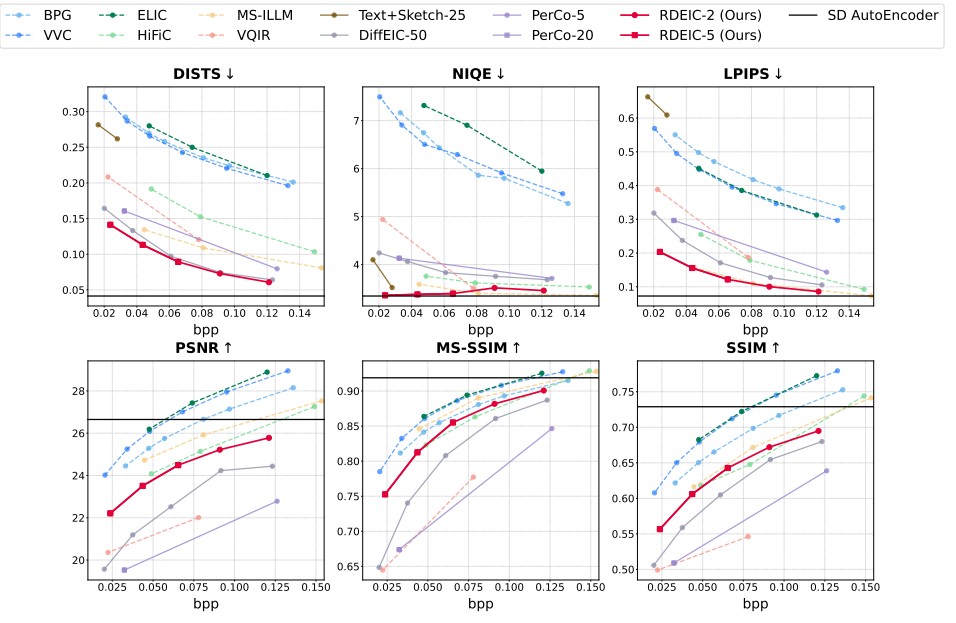

Figure 15: Quantitative comparisons with state-of-the-art methods on the Kodak dataset.

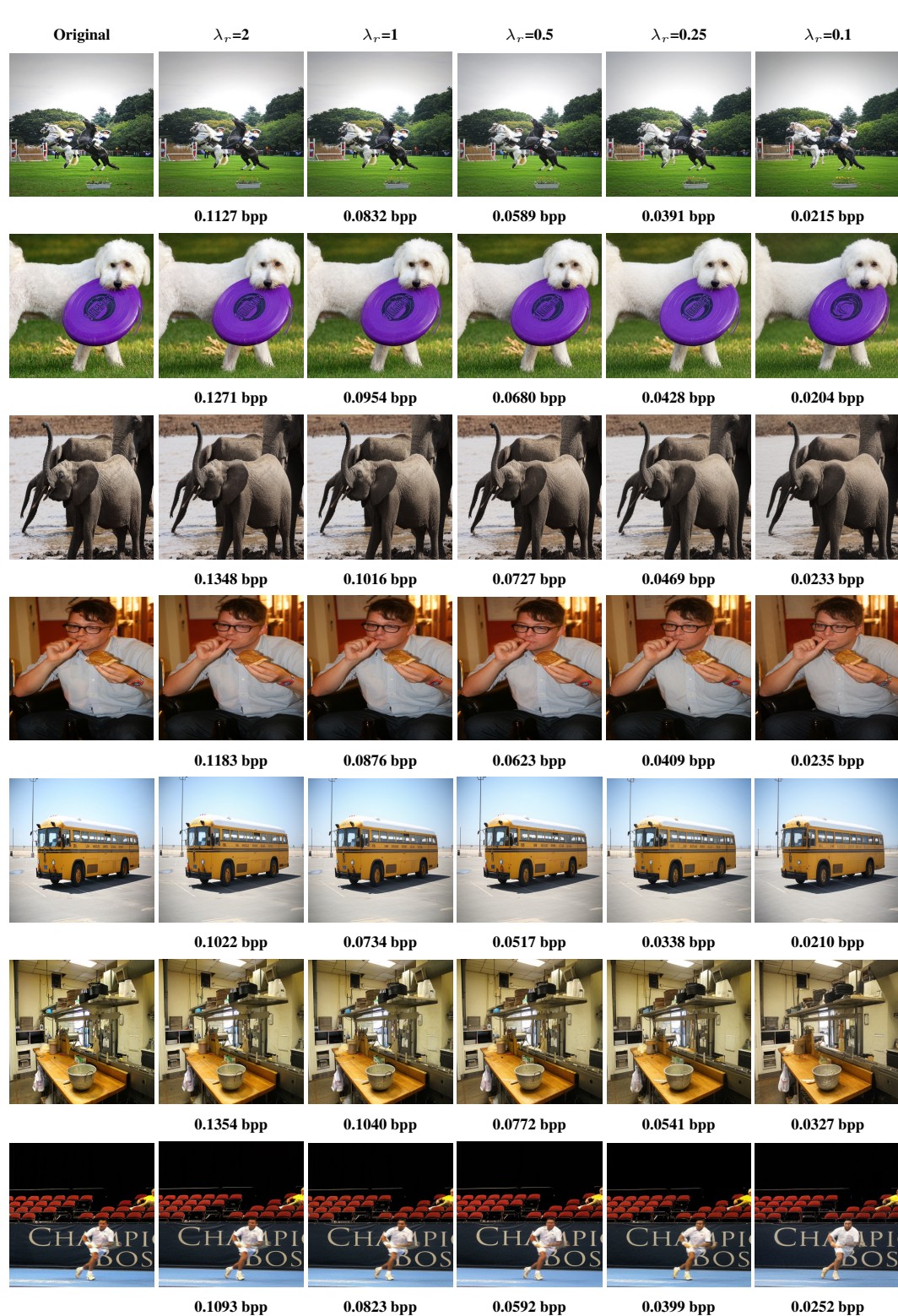

Figure 16: Visualization results of RDEIC on the MS-COCO 30k dataset at different bitrates.

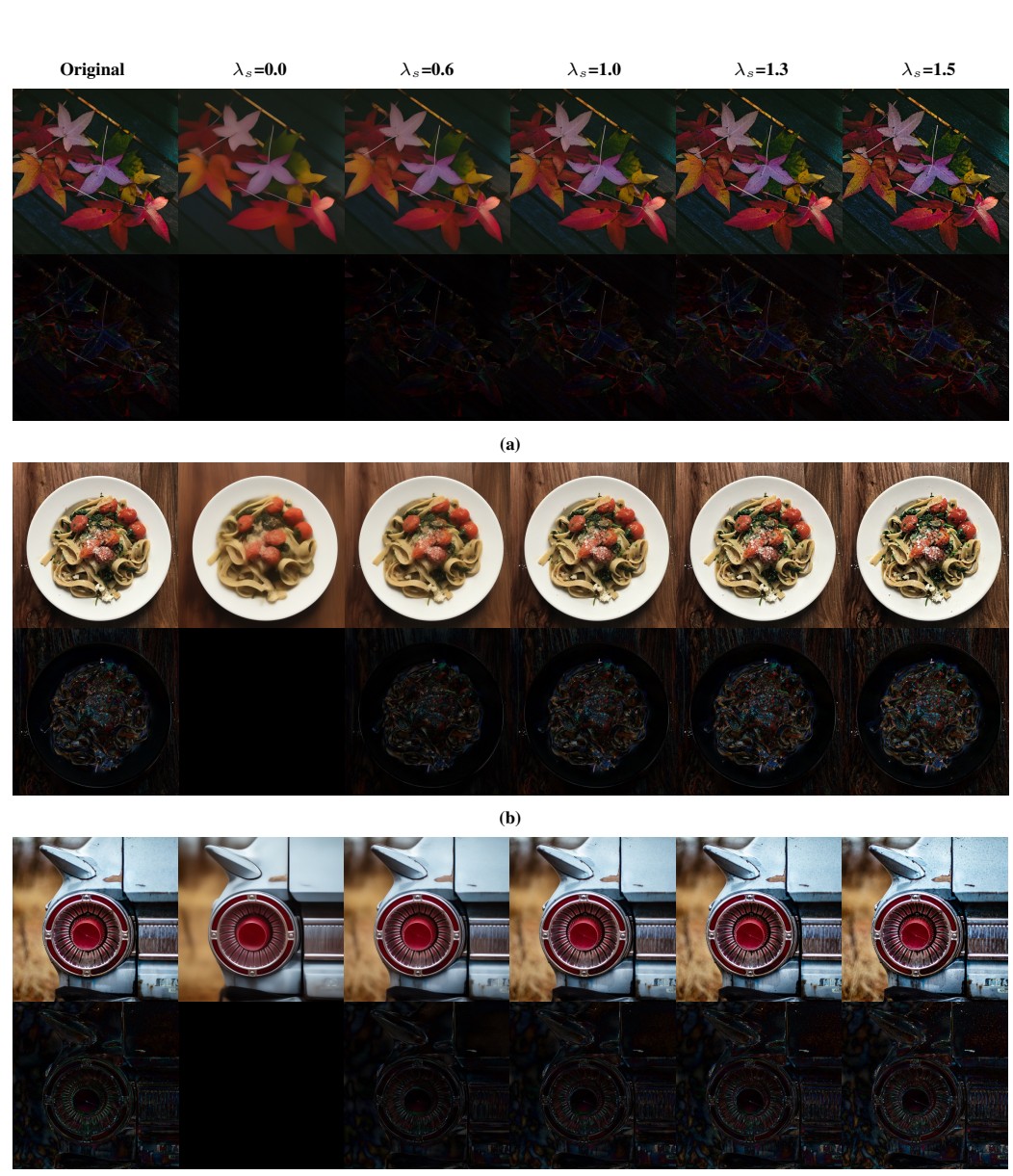

Figure 17: More results regarding the balance between smoothness and sharpness.

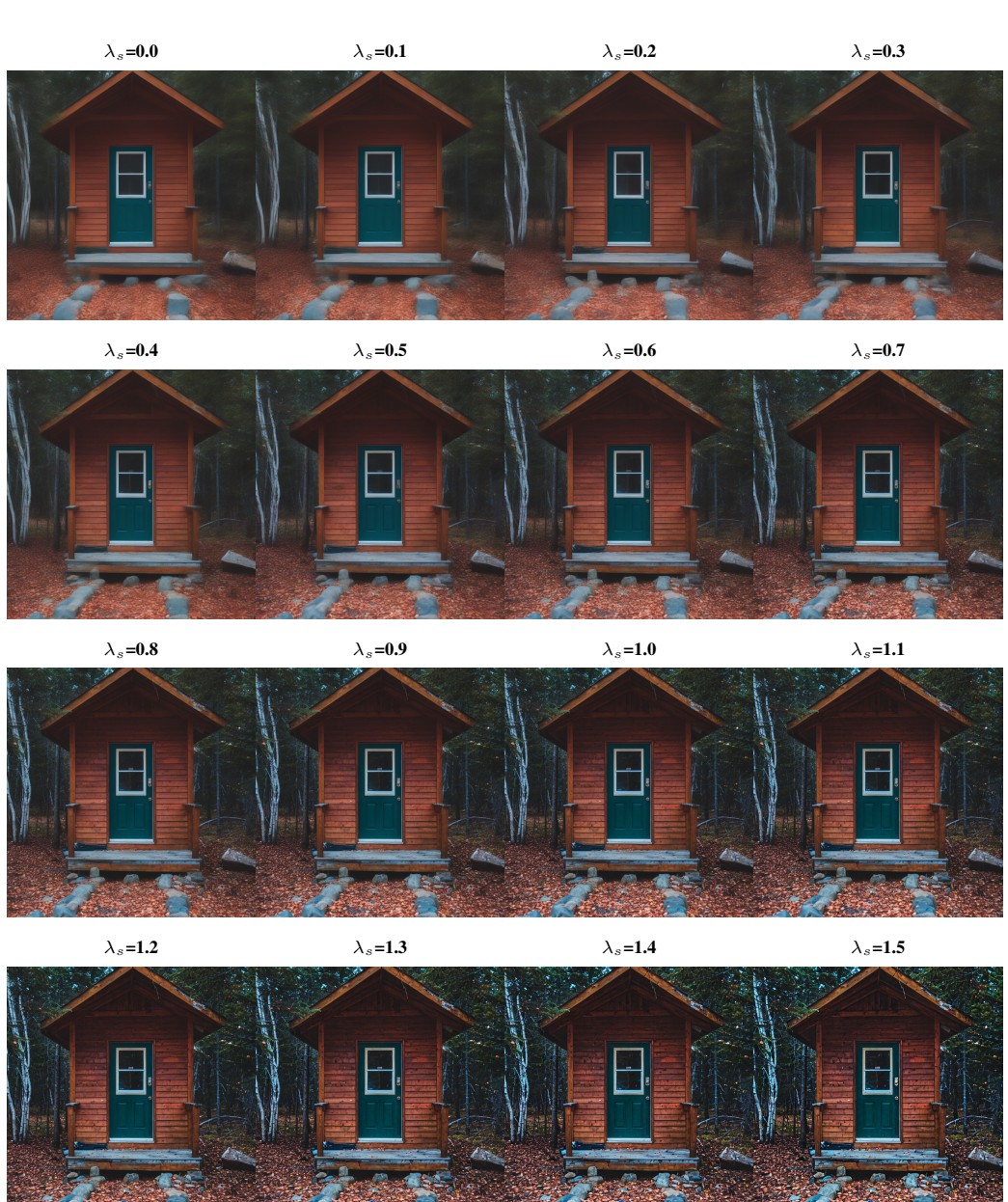

Figure 18: More results regarding the balance between smoothness and sharpness.

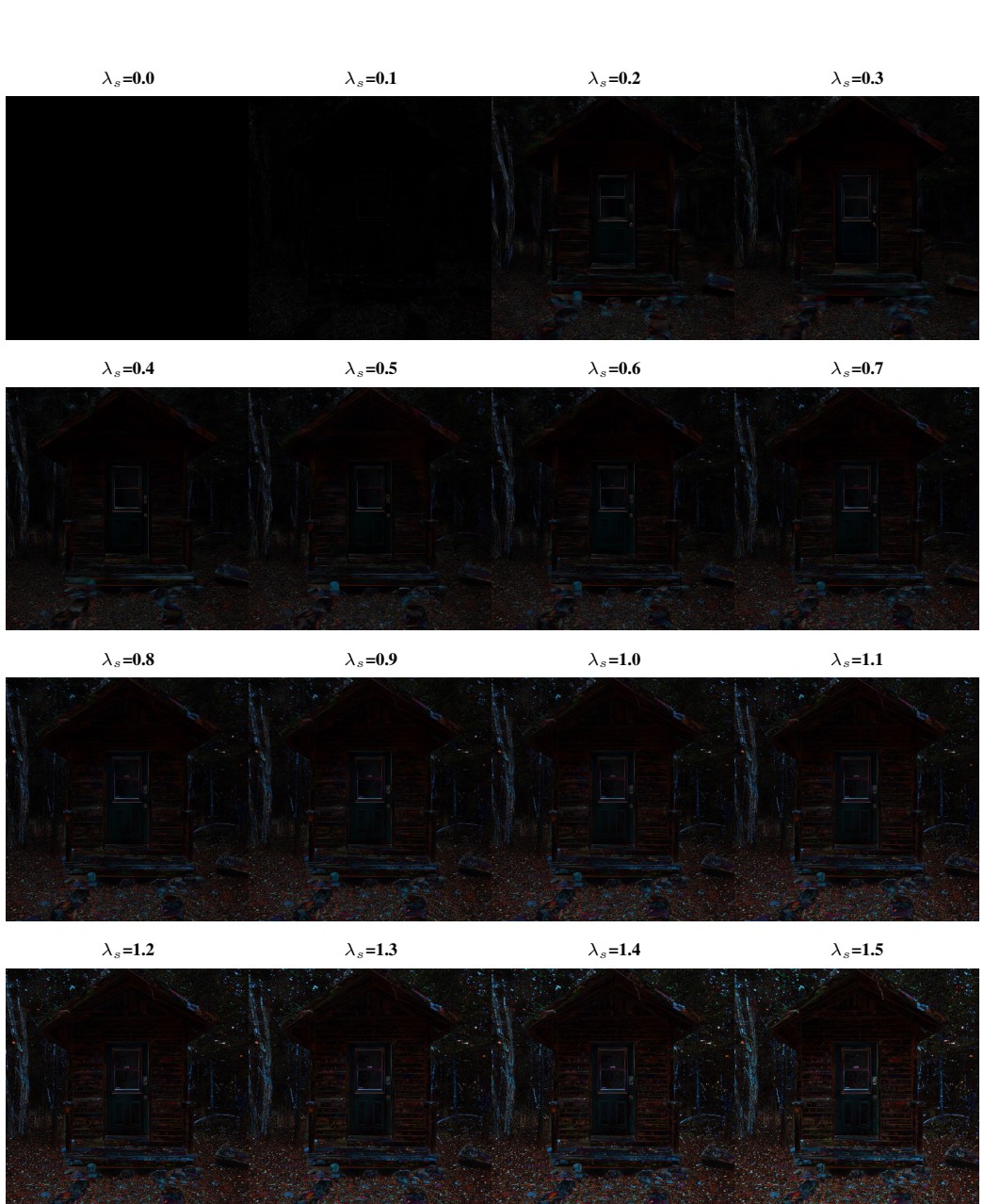

Figure 19: More results regarding the balance between smoothness and sharpness.

