# OpenReview forum: "Diffusion-based Extreme Image Compression with Compressed Feature Initialization"
_ICLR.cc/2025/Conference — ICLR 2025 Conference Withdrawn Submission_

### Official Review · Reviewer_X8Q6 · 2024-10-28

**Soundness:** 2
**Presentation:** 2
**Contribution:** 2
**Rating:** 5
**Confidence:** 4

**Summary:**

This paper introduces Relay Residual Diffusion Extreme Image Compression (RDEIC), a method for high-quality image compression at extremely low bitrates. RDEIC has three main components: (1) it begins denoising with compressed latent features plus noise instead of pure noise, reducing steps and improving fidelity; (2) it introduces a relay residual diffusion process, iteratively removing noise and residuals between compressed and target features, leveraging a pre-trained stable diffusion model for quality reconstruction; and (3) it applies a fixed-step fine-tuning strategy to minimize discrepancies between training and inference, further enhancing quality. Experimental results show that RDEIC achieves state-of-the-art visual quality, surpasses existing diffusion-based methods in fidelity and efficiency, and provides controllable detail generation to balance smoothness and sharpness.

**Strengths:**

1. Given the effectiveness and complexity of diffusion models, fast diffusion sampling as a practical research approach holds significant value and positively impacts the community.

2. The balance between smoothness and sharpness mentioned in the paper provides practical insights into this area. In a given compression state, determining how to map it to the sampling step
𝑁 can directly affect reconstruction quality. This mapping relationship is crucial to the model's effectiveness and stability, which the authors have explored in detail.

**Weaknesses:**

1. The novelty of this work is relatively modest, though it provides a valuable practical application in image compression. Many recent studies have explored similar approaches, starting the diffusion process from low-quality images rather than pure noise to enhance efficiency and accelerate sampling. Integrating degraded image embeddings into a pre-trained diffusion model as a plug-and-play module is also a relatively well-explored approach in the field of image processing. Prior works include:
   - [1] Lin, X., He, J., Chen, Z., Lyu, Z., Dai, B., Yu, F., ... & Dong, C. (2023) in "DiffBIR: Towards Blind Image Restoration with Generative Diffusion Prior",
   - [2] Wang, Y., Yu, Y., Yang, W., Guo, L., Chau, L. P., Kot, A. C., & Wen, B. (2023) in "ExposureDiffusion: Learning to Expose for Low-Light Image Enhancement" (ICCV),
   - [3] Ma, J., Zhu, Y., You, C., & Wang, B. (2023) in "Pre-trained Diffusion Models for Plug-and-Play Medical Image Enhancement" (MICCAI).

2. The Text-Sketch in Figure 1 and Figure 5 shows significant deviations in chroma reconstruction. I am unsure whether this is due to the baseline itself or if there was a mix-up between RGB and BGR channels during the experimental preprocessing stage. Additionally, the brightness of PerCo-20 in Figure 1 appears to be slightly biased compared to the ground truth. It is recommended to carefully examine the methods used for comparison, especially when the baselines are highly novel, and when results show noticeably unusual behavior, to ensure a fairer comparison.

3. Potential issue with variable control in the entropy model. The paper employs unusual entropy models (i.e., VQ-E and VQ-D) without adequate control or detailed explanation. This may lead to comparison results that do not accurately reflect the primary contribution of the proposed approach when contrasted with other algorithms, given that the precision of entropy models directly impacts compression efficiency and reconstruction quality.

4. Ambiguity in baseline selection. In Table 1 and Line 354, using “Ours” as the baseline results in a row of zeros, which may lead to ambiguity and does not align with traditional statistical practices (which typically use a control group as the baseline). It is advisable to clarify the baseline in the caption or table notes. Additionally, selecting a well-recognized baseline (e.g., JPEG, BPG, or a state-of-the-art compression method) for BD-rate comparison would provide a more straightforward understanding of the relative performance of each method.

5. Scoring issue with implementation versions. In Lines 442-443, the authors mention two implementation versions, yet both report a BD-rate of 0, which may cause confusion. It is recommended to provide a detailed explanation of the different implementations and clarify the reason for the BD-rate of 0 in each case.

6. Suggestions for improving formula clarity:
   - Clarity in the derivation from Eq.2 to Eq.4. The derivation from Eq.2 and Eq.3 to Eq.4 is crucial for the model’s structure but is not immediately clear. This derivation could directly impact the model's efficiency and accuracy. It is recommended to provide a more detailed explanation of these key steps in the main text to enhance understanding.

   - Ambiguity in the Definition of Eq.11. In traditional diffusion models (e.g., DDPM and Stable Diffusion), the noise estimator typically predicts total noise rather than noise at specific frequency bands. Interpreting $ \epsilon_{sd}(z_n, n) $ directly as a "low-frequency component" may lack theoretical support, especially without a clear basis for frequency division. The decomposition of predicted noise into low- and high-frequency components might be a heuristic approach, but further justification is needed to establish its rigor.

   - Undefined $ l_p $ in Eq.9. The definition of $l_p $ in Eq.9 is unclear. To improve understanding, it would be helpful for the authors to clearly specify the meaning of $ l_p $ and provide relevant context.

7. Minor formatting and typographical suggestions.
   - Line 100: Add commas before and after "i.e." for clarity.
   - Lines 220, 229, and 238: Add commas at the end of formulas to improve readability.

**Questions:**

1. Definition of "Extremely Low Bitrates". The standard for "extremely low bitrates" lacks a precise definition. Given varying content distributions (scenes) and the amount of high-frequency details, might "extremely low" have different thresholds? How would one define this threshold? Could the authors discuss the broader application potential of encoding methods in bandwidth-constrained scenarios? Additionally, does diffusion lose its value in compression at higher and medium bitrates?

2. Codebook Details. The approach involving "vector-quantized latent image representations" is intriguing. Could the authors elaborate on the learning and training process of the codebook loss? Specifically, how is the codebook initialized, and what is the interaction between the codebook and $ l_p$?

3. Since the multi-step sampling mechanism in diffusion leads to increased computational complexity in decoding, would placing diffusion in the encoding part or within the hyperprior yield different conclusions regarding complexity?

4. Role of the diffusion mechanism. Is diffusion effective mainly as a post-processing module to enhance perceptual quality, or does it also contribute to compact representation? A deeper analysis of the role of diffusion in improving perceptual quality versus compact representation would be insightful.

---

> ### Author Response · Authors · 2024-11-24
> **Response to Reviewer X8Q6 (Part I)**
>
> Thank you for your time and constructive comments. We have revised the manuscript based on your comments and address the weaknesses and questions raised in your review below:
>
> ---
>
> **Response to Weakness 1**:
>
> While recent works have explored leveraging the robust generative capability of Stable Diffusion for better perceptual quality (e.g, DiffBIR  [1] and DiffEIC [2]) or using residual diffusion for acceleration (e.g., ExposureDiffusion [3] and Resshift [4]), none have attempted to combine Stable Diffusion with residual diffusion. To the best of our knowledge, we are the first to successfully integrate Stable Diffusion into a residual diffusion framework.
>
> ---
>
> **Response to Weakness 2**:
>
> Thank you for raising this concern. We have carefully reviewed our experimental setup and found no issues with the implementation. The observed color deviations in Text+Sketch are expected, as this method reconstructs images solely from sketches and semantic text. This behavior is consistent with the results reported in its original paper. Additionally, the slight brightness bias observed in PerCo-20 is consistent with the results shown in the PerCo paper.
>
> ---
>
> **Response to Weakness 3**:
>
> Thank you for highlighting this concern. To address it, we have compared our baseline with DiffEIC. Note that the only difference between the two lies in the compression module. As shown in Fig. 6 and Table 2 (left) of the revised manuscript, the negligible differences in performance between DiffEIC and our baseline demonstrate that the choice of entropy model has minimal impact on overall performance. This confirms that the improvements in our method are primarily attributed to the proposed Relay Residual Diffusion (RRD) and Fixed-Step Fine-Tuning (FSFT) strategies, as further validated by the ablation study presented in Fig. 6 and Table 2 (left).
>
> >Table 2(left): The impact of RRD and FSFT on performance. Performance is represented by BD-rate (\%), using DiffEIC as the anchor. Distortion metrics include PSNR, MS-SSIM, and SSIM. Perceptual metrics include DISTS, FID, KID, NIQE, and LPIPS. DS denotes the number of denoising steps. 2/5 denotes that we use 2 denoising steps for the two models with larger bpp and 5 steps for the remaining models.
> | Methods        | DS    | Distortion | Perception | Average |
> |----------------|-------|------------:|------------:|---------:|
> | Baseline       | 50    | 7.4        | -1.8       | 2.8     |
> | +RRD           | 2/5   | -31.0      | 12.7       | -9.1    |
> | +RRD+FSFT      | 2/5   | -42.2      | -36.6      | -39.4   |
>
> ---
>
> **Response to Weakness 4**:
>
> Thank you for this valuable comment. Following your suggestion, we have reselected DiffEIC [2] as the anchor and added detailed notes to the table captions, as shown in Table 2 and Table 3 of the revised manuscript.
>
> >Table 3: BD-rate (\%) for different methods on the CLIC2020 dataset with DiffEIC as the anchor. For distortion-oriented methods (i.e., BPG, VVC, and ELIC), we omit their perceptual metrics. The best results are highlighted in **bold**.
> | Methods | | | Perception | | | | Distortion | | Average |
> |---|---:|---:|---:|---:|---:|---:|---:|---:|---:|
> |            | DISTS | FID | KID | NIQE | LPIPS | PSNR | MS-SSIM    | SSIM | |
> | BPG        | - | - | - | - | - | -66.2 | -32.8 | -40.3 | - |
> | VVC        | - | - | - | - | - | -77.8 | -51.3 | -58.6 | - |
> | ELIC       | - | - | - | - | - | **-82.7** | **-54.6** | **66.7** | - |
> | HiFiC      | 201.8 | 248.2 | 372.6 | -28.7 | 63.4 | -29.1 | 2.7 | 14.7 | 105.7 |
> | VQIR       | 71.8 | 183.9 | 156.7 | 32.4 | 51.3 | 16.4 | 43.9 | 57.8 | 76.8 |
> | PerCo      | 66.1 | 67.6 | 65.1 | 5.2 | 67.7 | 33.9 | 69.2 | 77.7 | 56.6 |
> | MS-ILLM    | 28.5 | 40.9 | 34.6 | **-85.4**| **-44.7**| -75.4 | -44.7 | -38.5 | -21.5 |
> | RDEIC (Ours)| **-17.9** | **-18.3**| **-22.1**| -83.7| -40.8| -61.3 | -32.7| -32.7| **-38.7** |
>
> ---
>
> **Reference**
>
> [1] Xinqi Lin, Jingwen He, Ziyan Chen, Zhaoyang Lyu, Ben Fei, Bo Dai, Wanli Ouyang, Yu Qiao, and Chao Dong. Diffbir: Towards blind image restoration with generative diffusion prior. arXiv preprint arXiv:2308.15070, 2023.
>
> [2] Zhiyuan Li, Yanhui Zhou, Hao Wei, Chenyang Ge, and Jingwen Jiang. Towards extreme imagecompression with latent feature guidance and diffusion prior. IEEE Transactions on Circuits and Systems for Video Technology, 2024.
>
> [3] Yufei Wang, Yi Yu, Wenhan Yang, Lanqing Guo, Lap-Pui Chau, Alex C Kot and Bihan Wen. Exposurediffusion: Learning to expose for low-light image enhancement. Proceedings of the IEEE/CVF International Conference on Computer Vision, 2023.
>
> [4] Zongsheng Yue, Jianyi Wang, and Chen Change Loy. Resshift: Efficient diffusion model for image super-resolution by residual shifting. In Thirty-seventh Conference on Neural Information Processing Systems, 2023

---

> > ### Author Response · Authors · 2024-11-24
> > **Response to Reviewer X8Q6 (Part II)**
> >
> > **Response to Weakness 5**:
> >
> > Thanks for pointing out this issue. For the two models corresponding to larger bpp values, we use 2 denoising steps, while for the remaining three models, we use 5 denoising steps. As shown in Fig. 4 of the revised manuscript, the performance points of these five models with different denoising steps collectively form the performance curve, serving as the anchor for calculating the BD-rate. To avoid unnecessary confusion, we have removed this column from Table 1 of the revised manuscript.
> >
> > ---
> >
> > **Response to Weakness 6**:
> >
> > 1. We have provided the detailed derivation from Eq. (2) to Eq. (4) in Appendix A.
> >
> > 2. We acknowledge that it is inaccurate to interpret $\epsilon_{sd}(z_n, c)$ as the low-frequency component of the noise itself. To enhance clarity and rigor, we have revised the manuscript to refer to $\epsilon_{sd}(z_n, c)$ as the “low-frequency control component.”
> >
> > 3. We have clarified in Sec. 3.1 that $\boldsymbol{l}_p$ represents the side information used in the image compression module, while $\hat{\boldsymbol{l}}_p$ refers to the vector-quantized result of $\boldsymbol{l}_p$, i.e., $\hat{\boldsymbol{l}}_p$ is the mapping of $\boldsymbol{l}_p$ to its closest codebook entry.
> >
> > ---
> >
> > **Response to Weakness 7**:
> >
> > Thanks a lot for pointing out these two issues, we have corrected them in the revised manuscript.

---

> > > ### Author Response · Authors · 2024-11-24
> > > **Response to Reviewer X8Q6 (Part III)**
> > >
> > > **Response to Question 1**:
> > >
> > > 1. **Definition of “Extremely Low Bitrates”**: In this work, we define “extremely low bitrates” as scenarios where the bits-per-pixel (bpp) falls below 0.1, aligning with the definition used in DiffEIC [1].
> > >
> > > 2. **Adjusting Thresholds Based on Content Complexity**: While it is theoretically possible to adjust the “extremely low” threshold based on content complexity, we argue that such adjustments are unnecessary. In academic evaluations, methods are assessed based on their average performance across diverse datasets, ensuring fairness and generality, regardless of individual image characteristics. Similarly, in practical applications, the primary focus is on meeting bandwidth or storage constraints, which are typically independent of image content.
> > >
> > > 3. **Broader Applications in Bandwidth-Constrained Scenarios**: The potential applications include satellite communication, underwater communication, and the transmission of non-critical internet images.
> > >
> > > 4. **Diffusion Models at Medium and High Bitrates**: Diffusion models retain significant value even at medium and high bitrates. Compared to GANs, diffusion models offer advantages such as greater training stability and superior generative performance. Methods like CDC [2] demonstrate the applicability of diffusion-based approaches in medium- and high-bitrate scenarios by leveraging their generative capabilities to produce high-quality, detail-rich reconstructions.
> > >
> > > ---
> > >
> > > **Response to Question 2**:
> > >
> > > 1. **Learning and Training Process**: Codebook provides a discrete quantization of the latent space, enabling efficient encoding of side information $l_p$ . The codebook loss is: $L_{cb} = \Vert sg(l_p) - \hat{l}_p \Vert_2^2 + \beta \Vert sg(\hat{l}_p) - l_p \Vert_2^2$, where  $sg(\cdot)$ is the stop-gradient operator and $\beta = 0.25$  in our experiments. $\Vert sg(\hat{l}_p) - l_p \Vert2^2$ ensures $l_p$ close to the nearest codeword $\hat{l}_p$ and $\Vert sg(l_p) - \hat{l}_p \Vert_2^2$ Encourages $\hat{l}_p$ moves closer to $l_p$. During training, the embeddings of codebook are dynamically updated based on the gradients from the codebook loss. In our implementation, we directly utilize CVQ-VAE [3]; further details can be found in the corresponding paper.
> > >
> > > 2. **Initialization**: The codebook is initialized uniformly as `self.embedding.weight.data.uniform_(-1.0 / n_e, 1.0 / n_e)`, where $n_e = 16384$ in our experiments.
> > >
> > > 3. **Interaction with  $l_p$**: Each element of  $l_p$  is replaced with its closest codeword $c_k$: $\hat{l}_q^{ij} = argmin _{c_k \in C} \Vert l_p^{ij} - c_k \Vert_2^2$, where C denotes the codebook, $I$ and $j$ denote positions.
> > >
> > > ---
> > >
> > > **Response to Question 3**:
> > >
> > > In compression tasks, placing diffusion in the encoding stage or within the hyperprior is not an optimal choice, as these stages cannot fully utilize the generative capabilities of diffusion models.
> > >
> > > The encoding stage primarily focuses on compactly representing the input data, and placing diffusion at this stage would shift computational complexity to the encoding end without contributing to the reconstruction process. Similarly, although placing diffusion within the hyperprior could theoretically reduce computational complexity because of the smaller feature resolution, this approach would mainly refine side information and have a relatively minor impact on the overall reconstruction quality.
> > >
> > > ---
> > >
> > > **Reference**
> > >
> > > [1] Zhiyuan Li, Yanhui Zhou, Hao Wei, Chenyang Ge, and Jingwen Jiang. Towards extreme imagecompression with latent feature guidance and diffusion prior. IEEE Transactions on Circuits and Systems for Video Technology, 2024.
> > >
> > > [2] Ruihan Yang and Stephan Mandt. Lossy image compression with conditional diffusion models. In Thirty-seventh Conference on Neural Information Processing Systems, 2023.
> > >
> > > [3] Chuanxia Zheng and Andrea Vedaldi. Online clustered codebook. In Proceedings of the IEEE/CVF International Conference on Computer Vision, pp. 22798–22807, 2023.

---

> > > > ### Author Response · Authors · 2024-11-24
> > > > **Response to Reviewer X8Q6 (Part IV)**
> > > >
> > > > **Response to Question 4**:
> > > >
> > > > Thank you for your insightful comment. Following your suggestion, we design two variants for comparison:
> > > >
> > > > 1. **W/o denoising process**: In this variant, the compression module is trained jointly with the noise estimator, but the denoising process is bypassed during the inference phase.
> > > >
> > > > 2. **W/o diffusion mechanism**: In this variant, the compression module is trained independently, completely excluding the influence of the diffusion mechanism.
> > > >
> > > > As shown in Fig. 10 of the revised manuscript, bypassing the denoising process results in significant degradation, particularly in perceptual quality. This highlights the critical role of the diffusion mechanism in enhancing perceptual quality during reconstruction. As shown in Fig. 11 of the revised manuscript, the diffusion mechanism effectively adds realistic and visually pleasing details.
> > > >
> > > > Additionally, Fig. 12(a) visualizes an example of bit allocation. The model trained jointly with the noise estimator allocates bits more efficiently, assigning fewer bits to flat regions (e.g., the sky in the image). Fig. 12(b) shows the cross-correlation between each spatial pixel in $(\boldsymbol{y} - \boldsymbol{\mu}) / \boldsymbol{\sigma}$ and its surrounding positions. The model trained jointly with the noise estimator exhibits lower cross-correlation, indicating reduced redundancy and more compact feature representations. These results indicate that the diffusion mechanism provides additional guidance for optimizing the compression module during training, enabling it to learn more efficient and compact feature representations.
> > > >
> > > > We have included a detailed discussion in Appendix C of the revised manuscript.

---

> ### Author Response · Authors · 2024-11-27
> **Look forward to your response**
>
> Dear Reviewer X8Q6,
>
> We hope you have had the opportunity to review our responses and clarifications. As the discussion period is nearing its conclusion, we would greatly appreciate it if you could confirm whether our updates have adequately addressed your concerns.
>
> Thank you for your time and consideration.
>
> Best regards,
>
> The Authors

---

### Official Review · Reviewer_PdRA · 2024-10-31

**Soundness:** 3
**Presentation:** 3
**Contribution:** 3
**Rating:** 6
**Confidence:** 4

**Summary:**

The paper presents a novel approach called Relay Residual Diffusion Extreme Image Compression (RDEIC), which improves upon traditional diffusion-based image compression methods. By leveraging compressed latent features and a residual diffusion process, RDEIC enhances fidelity and efficiency, addressing limitations of iterative denoising processes that typically begin with pure noise. Experimental results indicate significant performance gains in compression rates while maintaining image quality.

**Strengths:**

1) Introduces an innovative framework (RDEIC) that improves image compression efficiency.
2) Effectively addresses the fidelity issues present in existing diffusion-based methods.
3) Provides strong experimental results demonstrating the advantages of the proposed approach.

**Weaknesses:**

1) Limited discussion on the computational complexity of the new method.
2) Insufficient comparison with a broader range of existing compression techniques.
3) Potential overfitting concerns not addressed within the experimental analysis.

**Questions:**

1) Include a detailed analysis of the computational efficiency and resource requirements of RDEIC.
2) Expand the comparative analysis to include more baseline models and state-of-the-art techniques.
3) Address the possibility of overfitting by incorporating additional validation datasets or robustness tests.

---

> ### Author Response · Authors · 2024-11-24
> **Response to Reviewer PdRA**
>
> Thank you for your time and constructive comments. We have revised the manuscript based on your comments and address the weaknesses and questions raised in your review below:
>
> ---
>
> **Response to Weakness 1 & Question 1:**
>
> Following your suggestion, we have included additional ablation experiments on computational complexity. The proposed relay residual diffusion (RRD) framework enables image reconstruction with 5 or even 2 denoising steps, significantly improving the computational efficiency during decoding. As shown in Table 2 (right) of the revised manuscript, incorporating RRD reduces the denoising time by a factor of 10$\times$ to 25$\times$ compared to the baseline:
>
> > | Methods       | DS   | Denoising Time       | Speedup |
> |---------------|:------:|:----------------------:|---------:|
> | Baseline      | 50   | 4.349 ± 0.013        | 1×      |
> | +RRD          | 5    | 0.434 ± 0.002        | 10×     |
> |+RRD      | 2    | 0.173 ± 0.001        | 25×     |
>
> Additionally, the fixed-step fine-tuning strategy is purely a fine-tuning strategy and does not introduce any additional computational overhead during decoding. We have incorporated this discussion into the **Ablations** section.
>
> ---
>
> **Response to Weakness 2 & Question 2:**
>
> In the revised manuscript, we have expanded our comparative analysis by including additional baseline methods, namely the traditional compression standard BPG [1] and the VAE-based compression method ELIC [2]. These additions provide a broader context for evaluating the performance of our approach relative to both traditional and modern learning-based compression techniques.
>
> ---
>
> **Response to Weakness 3 & Question 3:**
>
> Thank you for raising this concern. To assess the robustness and generalization ability of RDEIC, we have conducted additional experiments on the larger MS-COCO 30k dataset, which comprises 30,000 images spanning a diverse range of categories and content types. This dataset was constructed by selecting the same images from the COCO2017 training set [3] as utilized in PerCo [4].
>
> As shown in Fig. 9 of the revised manuscript, RDEIC maintains consistent performance across this expanded dataset, demonstrating its ability to generalize effectively to unseen data, even in scenarios with more diverse and challenging content. Additionally, visualized examples of reconstructed images are provided in Fig. 16 of the revised manuscript to further illustrate the robustness of our approach.
>
> We have included this discussion in Appendix C of the revised manuscript.
>
> ---
>
> **Reference**
>
> [1] Fabrice Bellard. Bpg image format. 2014. URL https://bellard.org/bpg/
>
> [2] Dailan He, Ziming Yang, Weikun Peng, Rui Ma, Hongwei Qin, and Yan Wang. Elic: Efficient learned image compression with unevenly grouped space-channel contextual adaptive coding. In Proceedings of the IEEE/CVF Conference on Computer Vision and Pattern Recognition, pp.5718–5727, 2022.
>
> [3] Holger Caesar, Jasper Uijlings, and Vittorio Ferrari. Coco-stuff: Thing and stuff classes in context. In Proceedings of the IEEE conference on computer vision and pattern recognition, pp. 1209–1218, 2018.
>
> [4] Marlene Careil, Matthew J. Muckley, Jakob Verbeek, and Stéphane Lathuilière. Towards image compression with perfect realism at ultra-low bitrates. In The Twelfth International Conference on Learning Representations, 2024.

---

> > ### Comment · Reviewer_PdRA · 2024-11-27
> >
> > Thanks for your response. The author's response has answered my doubts.

---

> > > ### Author Response · Authors · 2024-11-28
> > > **Thanks to Reviewer PdRA**
> > >
> > > We sincerely appreciate your response and the valuable contribution your feedback has made to improving our work! If all your queries have been adequately addressed, we kindly ask you to consider raising your rating. However, if you still have any remaining doubts or concerns, we would be more than happy to engage in further discussion to clarify them.

---

> ### Author Response · Authors · 2024-11-27
> **Look forward to your response**
>
> Dear Reviewer PdRA,
>
> Thank you very much for your positive feedback on our work! We hope you have had the chance to review our responses and clarifications. As the discussion period is nearing its conclusion, we would greatly appreciate it if you could confirm whether our updates have fully addressed your concerns.
>
> Thank you again for your time and thoughtful review.
>
> Best regards,
>
> The Authors

---

### Official Review · Reviewer_Ejkn · 2024-11-02

**Soundness:** 2
**Presentation:** 2
**Contribution:** 2
**Rating:** 3
**Confidence:** 4

**Summary:**

This paper proposes Relay Residual Diffusion Extreme Image Compression method to achieve fidelity and efficiency. In particular, this paper use latent feature with added noise as the start point and employ residual diffusion to improve the fidelity. And this paper proposes a fixed-step fine-tuning strategy to reduce the number of steps.

**Strengths:**

This paper is clear in describing its contributions and methodology.
The experimental arrangement is relatively reasonable, and the ablation study can prove the effectiveness of the strategies proposed by the author.

**Weaknesses:**

The novelty is limited. Firstly, adding noise to the latent features is a common operation, which is used in many papers [1,2]. Secondly, the proposed residual diffusion is similar to ResShift[3]. The author should fully research the diffusion-based methods on low-level vision tasks published in the past two years and analyze the differences between them better.
[1] SeeSR: Towards Semantics-Aware Real-World Image Super-Resolution CVPR23
[2] Pixel-Aware Stable Diffusion for Realistic Image Super-Resolution and Personalized Stylization ECCV24
[3] ResShift: Efficient Diffusion Model for Image Super-resolution by Residual Shifting NIPS23

The motivation is not clear. In the third paragraph of Sec. 1, the author analysis the limitations of diffusion-based methods. The first limitation is ‘these methods rely on an iterative denoising process to reconstruct raw images from pure noise which is inefficient for inference’. The second limitation is ‘initiating the denoising process from pure noise introduces significant randomness, compromising the fidelity of the reconstructions.’ In addition to adding noise to the latent features, the author also employs a residual diffusion process and employ pre-trained stable diffusion to address these limitations. It is not clear remains unclear how residual diffusion and pre-trained stable diffusion can resolve the randomness caused by pure noise and improve the fidelity of the reconstructions.

There are two doubts about controllable detail generation. Firstly, the pre-trained stable diffusion is used to obtain low-frequency information. Since the pre-trained stable diffusion has not seen the inputs in the authors' task, why can it produce the expected results? Secondly, why did the authors choose to use pre-trained stable diffusion instead of directly using CFG?

**Questions:**

Figure 4 shows that the authors' method does not achieve the best results on metrics such as PSNR, MS-SSIM, and SSIM, and there is a significant gap compared to other methods. Noting that PSNR MS-SSIM, and SSIM are metrics used to evaluate fidelity. This is inconsistent with the authors' motivation. In the abstract, the authors mention that the proposed method aims to address the limitations of fidelity and efficiency.

The authors mention in their experiments that they trained five models, each corresponding to different λ_r values. However, in the comparative experiments (e,g, Tab.1, Tab. 2, Tab. 3, Fig. 4, Fig.5, Fig. 6, Fig.7, etc.), the authors do not specify which model's results were used. In addition, the author did not mention the guidance scale values used for these experimental results.

In Tab. 3, the author uses 2/5 in the DS column, so it is unclear whether the performance in the table refers to the 2-step model or the 5-step model. In addition, just using distortion of BD-rate or perception of BD-rate is not clear. The distortion includes PSNR, and SSIM, etc. and perception includes DISTS, FID, and LPIPS, etc. It is not clear which metrics distortion and perception represent respectively. The author should provide detailed results for metrics such as PSNR, SSIM, and LPIPS. Meanwhile, in the paper comparing the methods (PerCo, MS-ILLM), they did not use the bd-rate metric. Therefore, it is a good choice that the author just employs the values of PSNR, SSIM or LPIPS to demonstrate the performance and not use BD-rate.

In Tab. 2, the BD-rate of RDEIC with 2 DS is 0, while the BD-rate of RDEIC with 2 DS is also 0. So, which is the anchor in Tab. 2.

---

> ### Author Response · Authors · 2024-11-24
> **Response to Reviewer Ejkn (Part I)**
>
> Thank you for your time and constructive comments. We have revised the manuscript based on your comments and address the weaknesses and questions raised in your review below:
>
> ---
>
> **Response to Weakness 1**:
>
> Thank you for pointing out these relevant references. SeeSR [1] and PASD [2] embed the LR latent into the initial random noise at the terminal diffusion timestep N (1000) during inference, but still require numerous denoising steps for reconstruction (e.g., 50 in SeeSR and 20 in PASD). Resshift [3] constructs a Markov chain that transfers between degraded and target features by shifting the residual between them, substantially improving transition efficiency. However, ResShift's redesigned diffusion equation and noise schedule prevent it from leveraging the robust generative capability of pre-trained Stable Diffusion.
>
> In contrast, our RDEIC directly derives a novel residual diffusion equation from Stable Diffusion’s original diffusion equation, enabling seamless integration of per-trained Stable Diffusion to leverage its robust generative capability. To the best of our knowledge, this is the first successful integration of Stable Diffusion into a residual diffusion framework.
>
> We have incorporated this discussion into the **Related Work** section to clarify the novelty of our approach.
>
> ---
>
> **Response to Weakness 2**:
>
> Thanks for your insightful comment. Residual diffusion allows us to construct the starting point using a smaller timestep $N$ (300) instead of the terminal diffusion timestep $T$ (1000). As shown in Fig. 2(b), the resulting $\boldsymbol{z}_N$ retains most of the information from the compressed features $\boldsymbol{z}_c$, providing a strong foundation for detail generation. Additionally, starting from $N = 300$ naturally avoids the randomness and error accumulation associated with sampling from $n = 1000$ to $n = 300$.
>
> For Stable Diffusion, we leverage its robust generative capability to achieve high perceptual reconstruction at extremely low bitrates. It is important to clarify that Stable Diffusion is not used to resolve the randomness caused by pure noise or improve fidelity but rather to enhance the perceptual quality of the reconstruction.
>
> We have rephrased this content in the third paragraph of Sec. 1 to improve clarity and presentation.
>
> ---
>
> **Response to Weakness 3**:
>
> 1. For Stable Diffusion, we also start from $\boldsymbol{z}_N$ rather than pure noise. Since $\boldsymbol{z}_N$ retains most of the information from the compressed feature and no additional control conditions (e.g., text) are applied, the direct output of Stable Diffusion is low-frequency images lacking high-frequency details.
>
> 2. Our controllable detail generation method (Eq. (11)) aligns in form with classifier-free guidance (CFG). At each denoising step, the predicted noise is decomposed into two components, and the reconstruction is controlled by adjusting the guidance scale $\lambda_s$. In this framework, Stable Diffusion corresponds to the case where $\lambda_s = 0$.
>
> ---
>
> **Reference**
>
> [1] Rongyuan Wu, Tao Yang, Lingchen Sun, Zhengqiang Zhang, Shuai Li, and Lei Zhang. Seesr: Towards semantics-aware real-world image super-resolution. In Proceedings of the IEEE/CVF conference on computer vision and pattern recognition, pp. 25456–25467, 2024.
>
> [2] Tao Yang, Rongyuan Wu, Peiran Ren, Xuansong Xie, and Lei Zhang. Pixel-aware stable diffusion for realistic image super-resolution and personalized stylization. arXiv preprint arXiv:2308.14469, 2023.
>
> [3] Zongsheng Yue, Jianyi Wang, and Chen Change Loy. Resshift: Efficient diffusion model for image super-resolution by residual shifting. In Thirty-seventh Conference on Neural Information Processing Systems, 2023

---

> ### Author Response · Authors · 2024-11-24
> **Response to Reviewer Ejkn (Part II)**
>
> **Response to Question 1:**
>
> Diffusion-based extreme image compression methods are known for their exceptional performance in perceptual quality but often struggle to achieve high fidelity. As shown in Fig. 4 of the revised manuscript, diffusion-based approaches (solid lines) generally outperform other methods (dashed lines) in perceptual quality while exhibiting lower scores on fidelity metrics such as PSNR, MS-SSIM, and SSIM.
>
> Within this context, the proposed RDEIC achieves notable fidelity improvements compared to existing diffusion-based methods, such as DiffEIC [1] and PerCO [2]. These improvements align with our stated motivation to address the fidelity limitations of diffusion-based extreme image compression methods. While the fidelity of our RDEIC may not yet surpass that of traditional or other learning-based methods, it remains a meaningful step forward in improving fidelity of diffusion-based approaches.
>
> ---
>
> **Response to Question 2:**
>
> First, in image compression, it is common practice not to specify which model’s results are used when presenting comparative experiments, as each model corresponds to a different compression ratio (measured in bpp ). By default, results from all models are included to provide a comprehensive comparison. For instance, each point on the performance curves in Fig. 4 represents a model trained with a specific $\lambda_r$ value. Additionally, in Table 2 and Table 3 of the revised manuscript, the BD-rate (%) is calculated based on the performance of all models.
>
> Second, in all experiments, we set the guidance scale $\lambda_s=1$ by default unless otherwise specified. This clarification has been included in the revised manuscript.
>
> ---
>
> **Response to Question 3:**
>
> As stated in the clarification, 2/5 indicates that we use 2 denoising steps for the two models corresponding to larger bpp values and 5 steps for the remaining three models. The performance points from these five models collectively form the performance curve, serving as the anchor for comparison. In this table, distortion metrics include PSNR, MS-SSIM, and SSIM, while perceptual metrics include DISTS, FID, KID, NIQE, and LPIPS.
>
> Follow your suggestion, we have added detailed notes to the table caption to clarify this information and included the performance curves in Fig. 6 of the revised manuscript to provide a clearer demonstration of the results.
>
> > Table 2: The impact of RRD and FSFT on performance (left) and speed (right). Performance is represented by BD-rate (\%), using DiffEIC-50 as the anchor. Distortion metrics include PSNR, MS-SSIM, and SSIM. Perceptual metrics include DISTS, FID, KID, NIQE, and LPIPS. DS denotes the number of denoising steps. 2/5 denotes that we use 2 denoising steps for the two models with larger bpp and 5 steps for the remaining models. FSFT is a fine-tuning strategy that does not affect speed.
> | Methods       | DS   | Distortion | Perception | Average |           | Methods       | DS   | Denoising Time       | Speedup |
> |---------------|:------:|------------:|------------:|---------:|----------|---------------|:------:|:----------------------:|---------:|
> | Baseline      | 50   | 7.4        | -1.8       | 2.8     |             | Baseline      | 50   | 4.349 ± 0.013        | 1×      |
> | +RRD          | 2/5  | -31.0      | 12.7       | -9.1   |             | +RRD          | 5    | 0.434 ± 0.002        | 10×     |
> | +RRD+FSFT     | 2/5  | -42.2      | -36.6      | -39.4   |             |+RRD      | 2    | 0.173 ± 0.001        | 25×     |
>
> ---
>
> **Response to Question 4:**
>
> Thanks for pointing out this issue. For the two models corresponding to larger bpp values, we use 2 denoising steps, while for the remaining three models, we use 5 denoising steps. As shown in Fig. 4 of the revised manuscript, the performance points of these five models with different denoising steps collectively form the performance curve, serving as the anchor for calculating the BD-rate. To avoid unnecessary confusion, we have removed this column from Table 1 of the revised manuscript.
>
> ---
>
> **Reference**
>
> [1] Zhiyuan Li, Yanhui Zhou, Hao Wei, Chenyang Ge, and Jingwen Jiang. Towards extreme imagecompression with latent feature guidance and diffusion prior. IEEE Transactions on Circuits and Systems for Video Technology, 2024.
>
> [2] Marlene Careil, Matthew J. Muckley, Jakob Verbeek, and Stéphane Lathuilière. Towards image compression with perfect realism at ultra-low bitrates. In The Twelfth International Conference on Learning Representations, 2024.

---

> ### Author Response · Authors · 2024-11-27
> **Look forward to your response**
>
> Dear Reviewer Ejkn,
>
> We hope you have had the opportunity to review our responses and clarifications. We would be grateful if you could confirm whether our updates have fully addressed your concerns. Should you have any further comments or questions, we would be more than happy to address them at your convenience.
>
> Thank you once again for your valuable time and thoughtful feedback. We genuinely appreciate your efforts in reviewing our work.
>
> Best regards,
>
> The Authors

---

> > ### Comment · Reviewer_Ejkn · 2024-11-27
> >
> > Thanks for your response.
> >
> > It has resolved my confusion. However, there are still the following limitations.
> >
> > Firstly, compared to PASD and SeeSR, the innovation of modifying the start point is limited. Besides, the start points can also be adjusted, such as in AddNet [1], CCSR [2].
> >
> > Secondly, although ResShift trains a diffusion model from scratch, its core idea focuses on residual modeling. However, the idea of residual modeling in this paper is similar to that of ResShift. This paper just transfers residual modeling to stable diffusion. The innovation may not meet the requirements for ICLR.
> >
> > [1] https://arxiv.org/pdf/2404.01717
> > [2] https://arxiv.org/pdf/2401.00877v1

---

> ### Author Response · Authors · 2024-11-27
> **Response to Reviewer Ejkn regarding innovation**
>
> Thank you for your timely response and thoughtful feedback. We are glad to have clarified some of your concerns and would like to address the remaining points in detail.
>
> ---
>
> **1. Regarding Starting Point**
>
> First, after reviewing their papers and codebases, we found that the initial points in both methods are not adjustable as you mentioned. Specifically, in CCSR, the initial point is set to pure noise, as defined in its code:
>
>  `x_T = torch.randn(shape, device=model.device, dtype=torch.float32)`.
>
> AddNet, on the other hand, primarily focuses on controlling conditions during the denoising phase. Its initial point is similar to PASD and SeeSR, as reflected in its code:
>
> `parser.add_argument("--start_point", type=str, choices=['lr', 'noise'], default='lr')  # LR Embedding Strategy, choose 'lr latent + 999 steps noise' as diffusion start point.`
>
> Second, these previous methods only predict noise and do not consider the residual between degraded and target features. In contrast, our approach directly uses compressed features with slight noise as the starting point during training, enabling the network to simultaneously remove both noise and residual. This ensures consistency between the training and testing phases, improving both efficiency and reconstruction quality. We believe this makes our work novel compared to these prior methods.
>
> ---
>
> **2. Regarding similarities to ResShift**
>
> We acknowledge that our work shares a conceptual similarity with ResShift in utilizing residual modeling. However, we extend this idea by incorporating the powerful generative capability of pre-trained text-to-image diffusion models like Stable Diffusion into the residual diffusion framework—an area that has not been explored in previous works.
>
> It is important to note that Stable Diffusion is trained for noise prediction and cannot directly handle residuals. To leverage Stable Diffusion's robust generative capability while enabling the network to process residuals, we designed a new diffusion equation that satisfies the following conditions:
>
> - The diffusion equation retains the same structure as Stable Diffusion's equation, allowing seamless integration and utilization of its generative capability.
>
> - It can progressively add residuals in a manner analogous to noise addition.
>
> - $\boldsymbol{z}_N = \sqrt{\bar{\alpha}_N} \boldsymbol{z}_c + \sqrt{1-\bar{\alpha}_N}\epsilon_N$, where $\boldsymbol{z}_c$ is the compressed features.
>
> Specifically, we derived the following equation from Stable Diffusion’s diffusion framework:
>
> - $\boldsymbol{z}_{n} = \sqrt{\bar{\alpha}_n} (\boldsymbol{z}_0 + \lambda \frac{\sqrt{1-\bar{\alpha}_n}}{\sqrt{\bar{\alpha}_n}} \boldsymbol{e}) + \sqrt{1-\bar{\alpha}_n} \epsilon_n = \sqrt{\bar{\alpha}_n} \boldsymbol{z}_0 + \sqrt{1-\bar{\alpha}_n} (\lambda \boldsymbol{e} + \epsilon_n)$, where $\lambda=\frac{\sqrt{\bar{\alpha}_N}}{\sqrt{1-\bar{\alpha}_N}}$.
>
> This innovation allows the network to combine residual modeling with the robust generative capabilities of Stable Diffusion. Therefore, summarizing our work as “just transferring residual modeling to Stable Diffusion” does not fully capture the novelty and depth of our contributions.
>
> Furthermore, while our implementation leverages Stable Diffusion, the underlying methodology and derivation are general and can be readily extended to other text-to-image diffusion models.
>
> ---
>
> **Our Perspective on Innovation**
>
> Finally, we believe that discussing innovation without considering the specific task is inappropriate. In this work, our goal is to introduce residual diffusion to overcome the efficiency and fidelity limitations of existing diffusion-based extreme image compression methods. During this process, we addressed several critical challenges, including:
>
> - Integrating residual diffusion with pre-trained text-to-image diffusion models.
>
> - Achieving end-to-end training of the compression module within a residual diffusion framework.
>
> - Resolving inconsistencies between timestep-independent training and inference.
>
> - Enabling diverse reconstruction outputs without compromising efficiency.
>
> Experimental results demonstrate that our method achieves significant improvements in both reconstruction performance and computational efficiency. We believe that, in the field of image compression, our work is novel and makes meaningful contributions.
>
> ---
>
> We hope this response addresses your concerns and clarifies the novelty and contributions of our work. Thank you again for your valuable feedback.

---

### Official Review · Reviewer_1h9J · 2024-11-03

**Soundness:** 2
**Presentation:** 3
**Contribution:** 2
**Rating:** 3
**Confidence:** 4

**Summary:**

This paper introduces RDEIC, a novel diffusion model for extreme image compression that accelerates the denoising process through compression feature initialization. It draws on techniques from several papers, e.g., the codec framework scheme in GLC[1], and the control net in deffeic[2].The results provide evidence that the proposed scheme achieves SOTA performance.

**Strengths:**

1.	The paper is well-written and easy to follow, and the experiments are detailed and comprehensive.
2.	This paper reduces computational complexity by reducing the denoising step, which is valuable for resource-constrained environments.

**Weaknesses:**

1.	The paper has limited innovation. Its pipeline looks like a simple combination of GLC[1] and deffeic[2], utilizing the codec framework of GLC[1] and the generative model of deffeic[2]. However, the paper does not compare performance with GLC[1].
2.	This paper adopts a better performance generative model RDD instead of stable diffusion, and with the stronger generative ability of RDD, better performance is obtained. So if DiffEIC-50 also adopts RRD, will it achieve better performance?
3.	The conclusions of some visualization experiments are not rigorous enough. For example, in Fig. 1, despite the obvious subjective quality improvement of RDEIC, its bit rate is 7.5% higher than deffeic[2]. A similar problem can be observed in Figure 5.
4.	Some analysis needs to be included to show why RDEIC is worse than MS-ILLM on the NIQE metric.

[1] Jia Z, Li J, Li B, et al. Generative Latent Coding for Ultra-Low Bitrate Image Compression. CVPR 2024.

[2] Zhiyuan Li, Yanhui Zhou, Hao Wei, Chenyang Ge, and Jingwen Jiang. Towards extreme imagecompression with latent feature guidance and diffusion prior. IEEE Transactions on Circuits and Systems for Video Technology, 2024.

**Questions:**

See weakness.

---

> ### Author Response · Authors · 2024-11-24
> **Response to Reviewer 1h9J**
>
> Thank you for your time and constructive comments. We have revised the manuscript based on your comments and address the weaknesses and questions raised in your review below:
>
> ---
>
> **Response to Weakness 1**:
>
> As stated in the clarification, our focus is on improving the diffusion process and training strategy rather than proposing a novel pipeline. In summary, our primary innovation are as follows:
>
> - **Relay Residual Diffusion** : We propose a relay residual diffusion that effectively combines the efficiency of residual diffusion with the powerful generation capability of Stable Diffusion. **To the best of our knowledge, we are the first to successfully integrate Stable Diffusion into a residual diffusion framework. **
>
> - **Fixed-Step Fine-Tuning**: We design a fixed-step fine-tuning strategy that eliminates the discrepancy between training and inference, significantly enhancing reconstruction performance.
>
> - **Controllable Detail Generation**: We introduce a controllable detail generation method that enables a trade-off between smoothness and sharpness, allowing users to adjust the reconstruction results according to their preferences.
>
> Given these contributions, we believe it is inappropriate to assess the innovation of this work solely based on the pipeline structure. Regarding comparisons with GLC, we regret that we could not include results due to the unavailability of GLC’s official code. However, our comparisons with PerCo, DiffEIC, and other methods are sufficient to demonstrate the superiority of our proposed RDEIC.
>
> ---
> **Response to Weakness 2**:
>
> Thanks for your comment. RDD is not mentioned in this paper; did you mean RRD?
>
> Our RDEIC uses the same generative prior (Stable Diffusion) as DiffEIC. RRD is not a ``better generative model'' but rather the novel relay residual diffusion framework proposed in this paper, which combines the efficiency of residual diffusion with the robust generative capability of Stable Diffusion. Referring to Table 2(left) of the revised manuscript, applying the proposed RRD to DiffEIC would indeed improve its performance, as the difference between DiffEIC and our Baseline lies only in the compression module.
>
> >Table 2(left): The impact of RRD and FSFT on performance. Performance is represented by BD-rate (\%), using DiffEIC as the anchor. Distortion metrics include PSNR, MS-SSIM, and SSIM. Perceptual metrics include DISTS, FID, KID, NIQE, and LPIPS. DS denotes the number of denoising steps. 2/5 denotes that we use 2 denoising steps for the two models with larger bpp and 5 steps for the remaining models.
> | Methods        | DS    | Distortion | Perception | Average |
> |----------------|-------|------------:|------------:|---------:|
> | Baseline       | 50    | 7.4        | -1.8       | 2.8     |
> | +RRD           | 2/5   | -31.0      | 12.7       | -9.1    |
> | +RRD+FSFT      | 2/5   | -42.2      | -36.6      | -39.4   |
>
> ---
> **Response to Weakness 3**:
>
> Thank you for you insightful comment. In the revised manuscript, we have selected more appropriate visualization results in Fig. 1 and Fig. 5 to better illustrate the advantages of RDEIC.
>
> ---
> **Response to Weakness 4**:
>
> In the revised manuscript, we include the reconstruction performance of the SD autoencoder in Fig. 4 (indicated by the black horizontal line), which represents the theoretical upper limit of RDEIC’s performance. For bpp $>$ 0.06, the SD autoencoder performs worse than MS-ILLM in terms of NIQE, which explains why RDEIC also underperforms MS-ILLM on this metric.
>
> Additionally, as NIQE is a no-reference metric, a lower NIQE score does not always indicate better reconstruction quality in the context of extreme image compression. For instance, Text+Sketch achieves the best NIQE score but produces reconstructions that significantly deviate from the original image.

---

> > ### Comment · Reviewer_1h9J · 2024-11-25
> >
> > Thanks for your response. I still have the following concerns:
> >
> > Regarding response to weakness 1:
> >
> > -- According to the third innovation you gave, DiffEIC can also control the details of content generation, please elaborate on the difference.
> >
> > -- RDEIC almost duplicates GLC's encoder structure, which is another difference from DiffEIC. In order to fairly compare performance with DiffEIC, the performance gain from this change needs to be given.
> >
> > Regarding response to weakness 2:
> >
> > --Why is there a loss on the perception when only RRD is used?
> >
> > --It is recommended to add the performance of the RDEIC to the table so that the performance gains due to the inconsistency of the encoder can be seen.

---

> > > ### Author Response · Authors · 2024-11-25
> > > **Response to Reviewer 1h9J**
> > >
> > > Thanks for your prompt response! We address your concerns accordingly.
> > >
> > > ---
> > >
> > > _According to the third innovation you gave, DiffEIC can also control the details of content generation, please elaborate on the difference._
> > >
> > > DiffEIC effectively controls the details of content generation by adjusting the number of denoising steps. However, it requires a large number of denoising steps (e.g., 50 steps) to achieve reconstructions with more details, which significantly increases computational cost.
> > >
> > > In contrast, the proposed controllable detail generation method allows for controlling the details of content generation without increasing the number of denoising steps, providing a more flexible and efficient approach. Specifically, at each denoising step, the predicted noise is decomposed into a low-frequency control component and a high-frequency control component, and we control the details of content generation by adjusting the intensity of the high-frequency control component.
> > >
> > > ---
> > >
> > > _RDEIC almost duplicates GLC's encoder structure, which is another difference from DiffEIC. In order to fairly compare performance with DiffEIC, the performance gain from this change needs to be given._
> > >
> > > Thank you for highlighting this concern. To address it, we have compared our baseline with DiffEIC. Note that the only difference between the two lies in the compression module. As shown in Fig. 6 and Table 2 (left) of the revised manuscript, the negligible differences in performance between DiffEIC and our baseline demonstrate that the choice of compression module has minimal impact on overall performance. This confirms that the improvements in our method are primarily attributed to the proposed Relay Residual Diffusion (RRD) and Fixed-Step Fine-Tuning (FSFT) strategy, as further validated by the ablation study presented in Fig. 6 and Table 2 (left).
> > >
> > > ---
> > >
> > > _Why is there a loss on the perception when only RRD is used?_
> > >
> > > As shown in Fig. 7 of the revised manuscript, reducing the number of denoising steps inevitably leads to a decline in perceptual performance. With RRD, we use only 2 or 5 denoising steps, which is significantly fewer than the 50 steps used by the baseline. Therefore, the perceptual performance loss is expected and considered acceptable given the substantial improvement in efficiency.
> > >
> > > ---
> > >
> > > _It is recommended to add the performance of the RDEIC to the table so that the performance gains due to the inconsistency of the encoder can be seen._
> > >
> > > Thanks for your valuable comment.  "+RRD+FSFT" represents RDEIC in this table, which is the complete version of our proposed method. Are you referring to adding the performance of DiffEIC to the table for comparison? If so, we will revise the manuscript accordingly:
> > >
> > > >Table 2(left): The impact of RRD and FSFT on performance. Performance is represented by BD-rate (\%), using DiffEIC as the anchor. Distortion metrics include PSNR, MS-SSIM, and SSIM. Perceptual metrics include DISTS, FID, KID, NIQE, and LPIPS. DS denotes the number of denoising steps. 2/5 denotes that we use 2 denoising steps for the two models with larger bpp and 5 steps for the remaining models.
> > > | Methods        | DS    | Distortion | Perception | Average |
> > > |----------------|-------|------------:|------------:|---------:|
> > > | DiffEIC       | 50    | 0        |0       | 0     |
> > > | Baseline       | 50    | 7.4        | -1.8       | 2.8     |
> > > | +RRD           | 2/5   | -31.0      | 12.7       | -9.1    |
> > > | +RRD+FSFT      | 2/5   | -42.2      | -36.6      | -39.4   |
> > >
> > > It is evident that replacing the compression module with GLC does not result in performance improvements. Its primary contribution lies in slightly improving the encoding speed, as shown in Table 1 of the revised manuscript. This confirms that the enhancements in our method are primarily attributed to the proposed Relay Residual Diffusion (RRD) framework and Fixed-Step Fine-Tuning (FSFT) strategy.

---

> > > ### Author Response · Authors · 2024-11-27
> > > **Look forward to further discussion**
> > >
> > > Dear Reviewer 1h9J,
> > >
> > > Thank you once again for dedicating your valuable time to reviewing our paper and for your prompt feedback! We would greatly appreciate it if you could confirm whether our revisions have fully addressed your concerns. If you have any additional comments or questions, we would be more than happy to address them at your convenience.
> > >
> > > Best regards,
> > >
> > > The Authors

---

### Author Response · Authors · 2024-11-24
**Clarification of our RDEIC**

We thank all reviewers for their constructive feedback and the time they took to make the reviews. Before addressing your questions, we would like to clarify the focus of our work. In this work, **we aim to address fidelity and efficiency challenges commonly observed in existing diffusion-based extreme image compression methods (e.g., DiffEIC [1] and PerCo [2]) by improving the diffusion process and training strategy, rather than proposing novel network architectures.**

---

**Motivation**

As described in the introduction, we observed two major issues in existing diffusion-based extreme image compression methods:

- **Inefficient denoising process**: Existing diffusion-based extreme image compression methods follow the denoising process of DDPM [3], starting from pure noise to iteratively reconstruct the image. This requires a large number of denoising steps (e.g., 50 steps in DiffEIC) to achieve optimal reconstruction, making the process highly inefficient. Moreover, using random noise as the starting point introduces significant randomness, which compromises reconstruction fidelity.

- **Discrepancy between training and inference phases**: These methods train each time-step independently. For image compression, the lack of coordination among time-steps can result in error accumulation and suboptimal reconstruction.

---

**Methodology**

To address above issues, we propose the following solutions in this paper:

- **Compressed Feature Initialization**: Instead of starting from pure noise, we use the degraded feature $\boldsymbol{z}_c$ and slight noise $\epsilon_N$ to form the starting point $\boldsymbol{z}_N = \sqrt{\bar{\alpha}_N} \boldsymbol{z}_c + \sqrt{1-\bar{\alpha}_N}\epsilon_N$. As shown in Fig. 2(b), since $N$ (300) is much smaller than $T$ (1000), $\boldsymbol{z}_N$ retains most of the information from the compressed feature $\boldsymbol{z}_c$, providing a solid foundation for subsequent detail generation.

- **Relay Residual Diffusion (RRD)**: We also propose a novel relay residual diffusion to remove both added noise $\epsilon_N$ and the residual $\boldsymbol{e}$ ($\boldsymbol{e}=\boldsymbol{z}_c-\boldsymbol{z}_0$) contained in $\boldsymbol{z}_N$. The diffusion equation of our relay residual diffusion is directly derived from Stable Diffusion's diffusion equation, enabling seamless integration of pre-trained Stable Diffusion to leverage its robust generative capability for high perceptual reconstruction.**To the best of our knowledge, we are the first to integrate Stable Diffusion into a residual diffusion framework.**

- **Fixed-Step Fine-Tuning (FSFT)**: To eliminate the discrepancy between training and inference phases, we propose to fine-tune the model using the entire reconstruction process, further enhancing performance.

- **Controllable Detail Generation**: Inspired by classifier-free guidance (CFG), we propose a method to balance smoothness and sharpness, addressing the fixed-step constraint introduced by FSFT. This approach allows users to explore and customize outputs according to their personal preferences.

Equipped with the above components, the proposed RDEIC effectively combines the efficiency of residual diffusion with the powerful generation capability of Stable Diffusion, outperforming existing diffusion-based extreme image compression methods in both reconstruction performance and efficiency.

---

**Implement details**

We train five RDEIC models with different compression ratios, ranging from 0.02 bpp to 0.12 bpp. **During inference, we use 2 denoising steps for the two models with larger bpp and 5 denoising steps for the remaining three models.** Accordingly, we use 2/5 to represent the denoising steps of RDEIC.

---

We hope this clarification provides a better understanding of our motivation, methodology, and contributions. If you have any further questions, please do not hesitate to contact us.

**Reference**

[1] Zhiyuan Li, Yanhui Zhou, Hao Wei, Chenyang Ge, and Jingwen Jiang. Towards extreme imagecompression with latent feature guidance and diffusion prior. IEEE Transactions on Circuits and Systems for Video Technology, 2024.

[2] Marlene Careil, Matthew J. Muckley, Jakob Verbeek, and Stéphane Lathuilière. Towards image compression with perfect realism at ultra-low bitrates. In The Twelfth International Conference on Learning Representations, 2024.

[3] Jonathan Ho, Ajay Jain, and Pieter Abbeel. Denoising diffusion probabilistic models. Advances in neural information processing systems, 33:6840–6851, 2020.

---

### Note · Authors · 2025-01-25

I have read and agree with the venue's withdrawal policy on behalf of myself and my co-authors.